# High-Resolution Image Harmonization with Adaptive-Interval Color Transformation

**Quanling Meng**[1], **Qinglin Liu**[1], **Zonglin Li**[1], **Xiangyuan Lan**[2],
**Shengping Zhang**[1,2,*], **Liqiang Nie**[1]
[1]School of Computer Science and Technology, Harbin Institute of Technology, China
[2]Peng Cheng Laboratory, China
`quanling.meng@hit.edu.cn, qinglin.liu@outlook.com, zonglin.li@hit.edu.cn`
`lanxy@pcl.ac.cn, s.zhang@hit.edu.cn, nieliqiang@gmail.com`

## Abstract

Existing high-resolution image harmonization methods typically rely on global color adjustments or the upsampling of parameter maps. However, these methods ignore local variations, leading to inharmonious appearances. To address this problem, we propose an Adaptive-Interval Color Transformation method (AICT), which predicts pixel-wise color transformations and adaptively adjusts the sampling interval to model local non-linearities of the color transformation at high resolution. Specifically, a parameter network is first designed to generate multiple position-dependent 3-dimensional lookup tables (3D LUTs), which use the color and position of each pixel to perform pixel-wise color transformations. Then, to enhance local variations adaptively, we separate a color transform into a cascade of sub-transformations using two 3D LUTs to achieve the non-uniform sampling intervals of the color transform. Finally, a global consistent weight learning method is proposed to predict an image-level weight for each color transform, utilizing global information to enhance the overall harmony. Extensive experiments demonstrate that our AICT achieves state-of-the-art performance with a lightweight architecture. The code is available at `https://github.com/aipixel/AICT`.

## 1 Introduction

Image composition [3, 2, 24, 51] aims to combine a foreground object with a background image to create a realistic composite, which holds significant potential across various domains, including art, entertainment, commerce [3, 43, 52], and data augmentation [11, 30, 26]. However, since the foreground and background may be captured under different conditions, directly pasting the foreground onto the background usually results in an inconsistent appearance. To address this problem, image harmonization endeavors to adjust the color of the foreground to seamlessly integrate with the background, which plays a pivotal role in image editing.

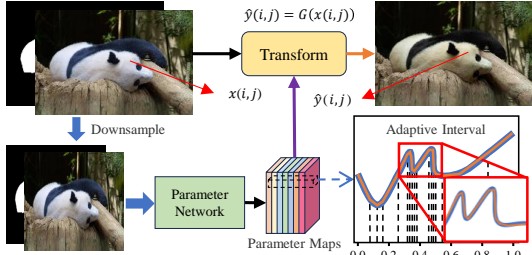

Figure 1: Our method predicts pixel-wise color transformations at low resolution and adaptively the adjusts sampling interval to model local non-linearities of the color transform at high resolution.

Traditional image harmonization methods [5, 19, 27, 28, 29, 36, 37] primarily concentrate on aligning the color statistics of the foreground to match the background using hand-crafted features. Since

---

*Corresponding author (s.zhang@hit.edu.cn).

38th Conference on Neural Information Processing Systems (NeurIPS 2024).

these methods lack consideration for the content of the composite images, they often yield suboptimal results when there are substantial differences in appearance between the foreground and background. With the rapid advance of deep learning, learning-based methods [38, 17, 9, 8, 22, 15, 33, 6, 44, 13] have become dominant and achieved remarkable progress. These methods usually adopt encoder-decoder based structures to learn the dense pixel-to-pixel transformation between composite images and ground-truth images at a low resolution (e.g., $256 \times 256$ pixels), while real-world applications increasingly demand high-resolution images. Although these methods can process images with any size theoretically, the computational cost required for high-resolution images is extremely expensive.

Recently, several methods [20, 21, 46, 12] have emerged to tackle the challenge of high-resolution image harmonization. To reduce computational costs, these methods usually use a low-resolution composite image to predict transformation parameters for processing the corresponding high-resolution composite image instead of directly generating the final image. These methods can be mainly categorized into two groups. Harmonizer [20] and S$^2$CRNet [21] focus on predicting image-level parameters to perform global color adjustments. However, these adjustments do not contain any semantic and local information, which leads to identical changes for pixels in different regions with the same color value. On the other hand, DCCF [46] and PCT-Net [12] predict low-resolution parameter maps and then directly upsample them to align with high-resolution composite images for pixel-wise color transformation. However, upsampling low-resolution parameter maps may introduce errors, which fails to model local non-linearities of the color transform at high resolution. In summary, these four methods ignore local color transformations across the foreground regions, which are prone to generate inharmony results in local regions.

To address this problem, we propose an Adaptive-Interval Color Transformation method (AICT) for high-resolution image harmonization. As shown in Figure 1, AICT predicts parameter maps for pixel-wise color transformations, rather than performing global color adjustments. Additionally, it adaptively adjusts the sampling interval to model local non-linearities of the color transformation at high resolution. To implement this complex transformation, we formulate the task as an image-based multiple curve estimation problem. Specifically, a parameter network is first proposed to generate multiple curves as position-dependent 3-dimensional lookup tables (3D LUTs), which use the color and position of each pixel to perform pixel-wise color transformations. Then, to enhance local variations adaptively, we propose an adaptive interval learning method, which separates a color transform into a cascade of sub-transformations using two position-dependent 3D LUTs to achieve the non-uniform sampling

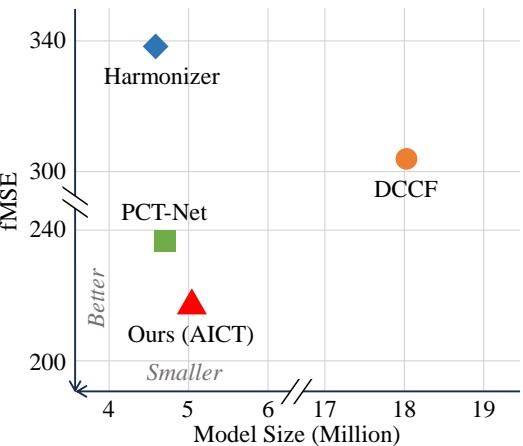

Figure 2: Model size vs. performance (fMSE score) comparison on the full-resolution images of the iHarmony4 dataset [8]. The proposed method achieves state-of-the-art performance while maintaining a lightweight architecture.

intervals of the color transform. Finally, a global consistent weight learning method is proposed to predict an image-level weight for each color transform, utilizing global information to enhance the overall harmony by modeling the influence of background brightness on the foreground. As shown in Figure 2, AICT achieves state-of-the-art performance in foreground-normalized MSE (fMSE) on the full-resolution images of the iHarmony4 dataset [8] while maintaining a model size comparable to PCT-Net [12]. Our main contributions are as follows:

- We propose an Adaptive-Interval Color Transformation method to perform pixel-wise color transformations and model local non-linearities of the color transformation for high-resolution image harmonization.

- We propose an adaptive interval learning method to achieve a flexible sampling point allocation and a global consistent weight method to utilize global information to enhance the overall harmony.

- Extensive experiments demonstrate that the proposed method achieves state-of-the-art performance in high-resolution image harmonization while maintaining a lightweight and simple network architecture.

## 2 Related Work

### 2.1 Image Harmonization

To reduce computational costs, existing high-resolution image harmonization methods [20, 21, 46, 12, 7, 40] usually takes a low-resolution image as input to predict transformation parameters for processing the corresponding high-resolution image instead of directly outputting the final image. For example, Harmonizer [20] employs a neural network to regress filter arguments based on low-resolution images, which are then used for several white-box filters to adjust various aspects of the foreground, including brightness, contrast, and other characteristics. $S^2$CRNet [21] focuses on extracting spatial-separated embeddings from low-resolution images to predict parameters of the piece-wise curve mapping for performing color-wise transformations. Similarly, Wang et al. [40] utilize down-sampled images to predict global RGB curves for performing color correction at higher resolutions. Furthermore, they propose to predict and upsample low-resolution shading maps to address local tonal variations. These three methods predict image-level parameters to perform global color adjustments. Unlike these methods, DCCF [46] processes low-resolution images to acquire human comprehensible neural filter maps, which are subsequently upsampled and applied to the original input image. PCT-Net [12] predicts affine transformation parameter maps based on low-resolution images and upsamples them to match high-resolution images. Both methods predict low-resolution parameter maps and then upsample them directly to align with high-resolution images for pixel-wise transformations. In addition, CDTNet [7] performs pixel-to-pixel transformation at low resolution and color-to-color transformation at high resolution in parallel. It subsequently utilizes a refinement module to integrate the two intermediate outputs. The above methods typically rely on global color adjustments or the upsampling of parameter maps, which ignore local variations. Our method performs pixel-wise color transformations and models local non-linearities for high-resolution image harmonization.

### 2.2 LUT-based Image Enhancement

A lookup table (LUT) defines a set of tables consisting of output values, where each value can be addressed by a set of indices. When provided with an input combination, it can output a corresponding value by performing lookup and interpolation operations, so it usually serves as an effective and efficient representation of a univariate or multivariate function. Zeng et al. [49] propose to learn multiple basis 3D LUTs and predict content-dependent weights to fuse them into an image-adaptive LUT for photo enhancement. To consider global scenarios and local spatial information, Wang et al. [41] introduce a lightweight two-head weight predictor for image-level scenario adaptation and pixel-wise category fusion. Yang et al. [47] achieve a more flexible sampling point allocation in 3D LUTs by adaptively learning the non-uniform sampling intervals in the 3D color space. To enhance the expressiveness of the LUT, Yang et al. [48] decompose a single color transform into a cascade of sub-transforms and use 1D LUTs to increase cell utilization within the 3D LUT. Zhang et al. [50] employ hash techniques to reduce the space complexity of 3D LUTs. Liu et al. [23] propose to learn a context map for the pixel-level category and a group of image-adaptive coefficients for achieving context-aware 4D LUT. These methods typically predict several weight parameters to fuse pre-trained LUTs for global RGB-to-RGB transformations, while our method predicts position-dependent 3D LUTs for pixel-wise color transformations.

## 3 Method

Given a composite image $\tilde{I} \in \mathbb{R}^{H \times W \times 3}$ and a binary mask $M \in \{0,1\}^{H \times W}$ that indicates the foreground region to be harmonized, image harmonization aims to adjust the color of the foreground region to obtain a harmonized image $\hat{I} \in \mathbb{R}^{H \times W \times 3}$ close to the ground truth image $I \in \mathbb{R}^{H \times W \times 3}$. To achieve high-resolution image harmonization at a low computational cost, we formulate the task as an image-based multiple curve estimation problem and propose an Adaptive-Interval Color Transformation method (AICT) as shown in Figure 3. AICT consists of a high resolution (HR)

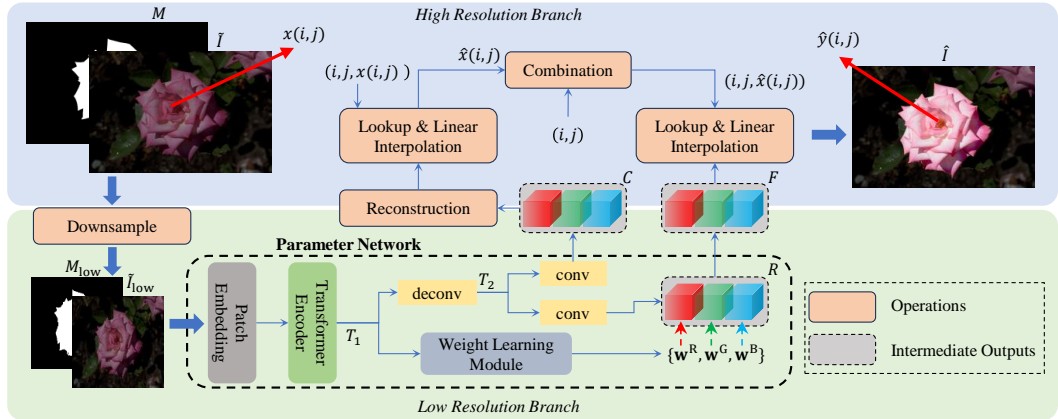

Figure 3: The framework of the proposed method. AICT consists of a high resolution (HR) branch and a low resolution (LR) branch. In the LR branch, the composite image $\tilde{I}$ and composite mask $M$ are downsampled to predict two parameter maps $C$ and $F$. In the HR branch, $C$ is used to redistribute the color values of $\tilde{I}$ into specific ranges of the following $F$, which achieves adaptively adjusting the sampling interval of the color transformation. Finally, the new color values and pixel coordinates are mapped to final color values using $F$.

branch and a low resolution (LR) branch. In the LR branch, $\tilde{I}$ and $M$ are first downsampled to obtain a low-resolution composite image $\tilde{I}_{\text{low}} \in \mathbb{R}^{H' \times W' \times 3}$ and foreground mask $M_{\text{low}} \in \{0, 1\}^{H' \times W'}$, where $H' < H$ and $W' < W$. They are then fed into a parameter network to predict two parameter maps $C \in \mathbb{R}^{H' \times W' \times Q}$ and $F \in \mathbb{R}^{H' \times W' \times Q}$ and they are regarded as curves in the form of position-dependent 3D LUTs, where $Q$ is equal to three times the number of knot points in a curve. In the HR branch, $C$ is used to process color values and pixel coordinates to redistribute the color values of $\tilde{I}$ into specific ranges of the following $F$ for achieving pixel-wise adaptive adjustment of the sampling interval. $F$ is used to map the redistributed color values and pixel coordinates to final color values, achieving pixel-wise color transformation. Therefore, the color transformation in our method includes two cascaded pixel-wise sub-transformations.

## 3.1 Adaptive Interval Learning

In the LR branch, $\tilde{I}_{\text{low}}$ and $M_{\text{low}}$ are fed into the parameter network to produce $C$ and $F$. As shown in Figure 3, we use the Transformer encoder [13] to construct the parameter network, which consists of multiple attention layers. Firstly, the low-resolution composite image $\tilde{I}_{\text{low}}$ and foreground mask $M_{\text{low}}$ are divided into patches, which are then projected into the embedding space. Positional encodings are added to the embedded patches, which are processed by the Transformer encoder. The output of the Transformer encoder is reassembled to obtain the feature $T_1 \in \mathbb{R}^{64 \times 64 \times 256}$, and a deconvolution layer is applied to reduce the feature channel number and perform upsampling to obtain the feature $T_2 \in \mathbb{R}^{H' \times W' \times 12}$. Subsequently, $T_2$ is processed through a $1 \times 1$ convolution to obtain the parameter map $C \in \mathbb{R}^{H' \times W' \times Q}$. We individually process each color channel and divide $C$ into three parameter maps $C^{\text{R}} \in \mathbb{R}^{H' \times W' \times K}$, $C^{\text{G}} \in \mathbb{R}^{H' \times W' \times K}$, and $C^{\text{B}} \in \mathbb{R}^{H' \times W' \times K}$, where $Q$ is equal to $3 \times K$. Taking the red channel of an RGB image as an example, the corresponding parameter map is $C^{\text{R}}$. To achieve adaptive interval learning, we need to reconstruct the parameter map $C^{\text{R}}$. The softmax function is used to obtain the normalized interval $V^{\text{R}} \in [0, 1]^{H' \times W' \times M} = \text{Softmax}(C^{\text{R}}, \text{axis} = 3)$, where $\text{axis} = 3$ indicates that normalization is performed along the third dimension [47] of $C^{\text{R}}$. We perform cumulative summation in the third dimension of $V^{\text{R}}$ and then add an origin for each position to obtain the sampling coordinates $K^{\text{R}} \in [0, 1]^{H' \times W' \times (M+1)}$, which can be expressed as $K^{\text{R}} = [Z; \text{Cumsum}(V^{\text{R}}, \text{axis} = 3)]$, where $Z$ is a $H' \times W' \times 1$ matrix filled with zero values, and the $[\cdot; \cdot]$ denotes the concatenation operation. In such a way, each value in $K^{\text{R}}$ is within the range of 0 to 1 and maintains the monotone increasing properties along the third dimension ($K^{\text{R}}_{c,b,i} < K^{\text{R}}_{c,b,j}$, for $c, b \in \mathbb{I}_0^{255}$, $i, j \in \mathbb{I}_0^M$, and $i < j$).

We use position-dependent 3D LUTs to achieve pixel-wise adaptive adjustment of the sampling interval. The position-dependent 3D LUT maps spatial coordinates and color values to new color values [35]. We treat $K^R$ as a position-dependent 3D LUT, which utilizes lookup and interpolation operations to serve as a color mapping curve. For a given coordinate $(i, j)$ and its color value $x(i, j)$ in the red channel of $\tilde{I}$, we can find the corresponding position $(u, v, w)$ in $K^R$, where $u = i\frac{H'-1}{H-1}$, $v = j\frac{W'-1}{W-1}$, $w = x(i, j)\frac{K-1}{C_{max}}$ and $C_{max}$ represents the maximum color value in the image. Based on $(u, v, w)$, we can obtain adjacent 8 sampling points in $K^R$ and perform trilinear interpolation to produce a new color value $\hat{x}(i, j)$. The entire process can be formulated as

$$\hat{x}(i, j) = t(l(i, j, x(i, j), K^R)) \tag{1}$$

where $t$ and $l$ denote trilinear interpolation and lookup operations, respectively. The upper section of Figure 4 demonstrates the color redistribution process. Here, each coordinate and its corresponding color value are mapped to a new color value using $K^R$. Similarly, we can obtain redistributed color values for the other color channels in the same way.

## 3.2 Pixel-Wise Color Transform

After obtaining the redistributed color values, we utilize the parameter map $F$ to achieve pixel-wise color transformation. As shown in Figure 3, we apply a $1 \times 1$ convolution to the feature $T_2$ for obtaining the intermediate parameter map $R \in \mathbb{R}^{H' \times W' \times Q}$, which are divided into three parameter maps $R^R \in \mathbb{R}^{H' \times W' \times K}$, $R^G \in \mathbb{R}^{H' \times W' \times K}$, and $R^B \in \mathbb{R}^{H' \times W' \times K}$ according to the color channel. When inserting the foreground into a new background, the average brightness of the foreground will be influenced by the background. For example, the inserted foreground may experience overall darkening, brightening, or a bias toward a certain color. As shown in Figure 5, the foreground region un-

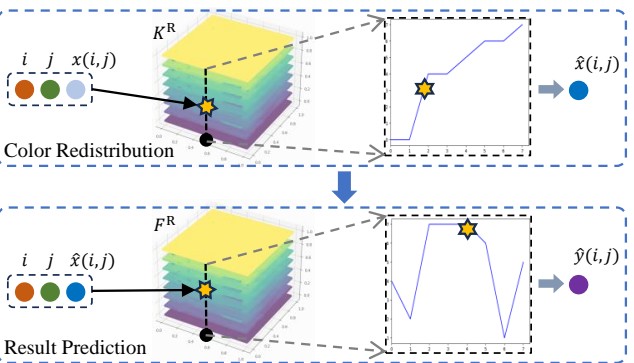

Figure 4: The illustration of the color redistribution and result prediction. The coordinates and color values are first mapped to new color values using $K^R$. Then, the final color values are obtained by using $F^R$ according to the coordinates and redistributed color values.

dergoes darkening after image harmonization. Therefore, we propose a globally consistent weight learning method and design a weight learning module to learn image-level parameters for controlling overall color transformations. These parameters can also be regarded as scene classification, which are adjusted based on different scenes. The weight learning module consists of two $3 \times 3$ convolutional layers, Batch Normalization (BN) layers [18], max-pooling layers, ReLU activation functions, and one fully-connected layer, making it lightweight. It processes $T_1$ to predict weight vectors $\mathbf{w}^R \in \mathbb{R}^K$, $\mathbf{w}^G \in \mathbb{R}^K$, and $\mathbf{w}^B \in \mathbb{R}^K$, which are used to multiply with $R^R$, $R^G$, and $R^B$ to obtain the final parameter maps $F^R \in \mathbb{R}^{H' \times W' \times K}$, $F^G \in \mathbb{R}^{H' \times W' \times K}$, and $F^B \in \mathbb{R}^{H' \times W' \times K}$, respectively.

We also treat $F^R$ as a position-dependent 3D LUT. For a given coordinate $(i, j)$ and its redistributed color value $\hat{x}(i, j)$ in the red channel of $\tilde{I}$, lookup and interpolation operators are performed in $F^R$ to produce the final output color value $\hat{y}(i, j)$. The bottom part of Figure 4 illustrates the result prediction process. Due to the difference in resolution, each pixel in the parameter maps $K^R$ and $F^R$ corresponds to a local region of the composite image. However, the color values in a

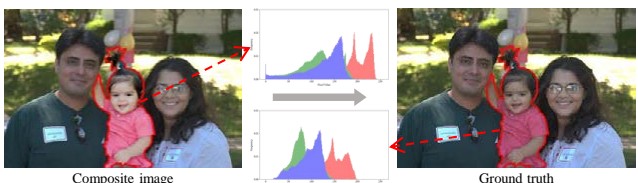

Composite image       Ground truth

Figure 5: The illustration of the overall impact of the background on the foreground. The foreground is outlined in red. The pixel histogram statistics indicate that the overall foreground region undergoes darkening after the process of image harmonization.

region are usually distributed within certain ranges. Therefore, $K^{\text{R}}$ aims to redistribute these color values across the entire color value range, thereby improving the utilization of sampling points in $F^{\text{R}}$ and enhancing the expressiveness of the curve. By combining these two curves ($K^{\text{R}}$ and $F^{\text{R}}$), our method achieves the capability of adaptively adjusting the sampling intervals. Finally, the result images corresponding to each channel are obtained by transforming the color values at each position of $\tilde{I}$, which are then concatenated to produce the harmonized image $\hat{I}$. As we process each color channel separately for an RGB image, the parameter network needs to predict six curves.

### 3.3 Loss Functions

During the training phase, the loss $\mathcal{L}_{\text{high}}$ is calculated in the HR branch based on the difference between $\hat{I}$ and $I$. To improve the performance of AICT for images with small foregrounds, we adopt the foreground-normalized MSE loss [33], which is formulated as

$$\mathcal{L}_{\text{high}} = \frac{\sum_{i=1}^{H} \sum_{j=1}^{W} \left\| \hat{I}_{i,j} - I_{i,j} \right\|_2^2}{\max\{A_{\min}, \sum_{i=1}^{H} \sum_{j=1}^{W} M_{i,j}\}} \tag{2}$$

where $A_{\min}$ is a hyper-parameter to stabilize training. By replacing $\tilde{I}$ with $\tilde{I}_{\text{low}}$ in the framework, we can also obtain the low-resolution harmonized image $\hat{I}_{\text{low}}$ and the loss $\mathcal{L}_{\text{low}}$. We also apply the foreground-normalized MSE loss to minimize the difference between $\hat{I}_{\text{low}}$ and the low-resolution version of the ground truth $I_{\text{low}}$. Overall, our network can be trained by optimizing the combination of the losses above

$$\mathcal{L} = \mathcal{L}_{\text{high}} + \lambda \mathcal{L}_{\text{low}} \tag{3}$$

where $\lambda$ is a hyper-parameter that controls the weight of $\mathcal{L}_{\text{low}}$. Our parameter network is trained in an end-to-end manner.

## 4 Experiments

### 4.1 Experimental Setting

#### 4.1.1 Dataset and Evaluation Metrics

Following the same settings as previous methods [20, 46, 12], we train and evaluate our method on the iHarmony4 dataset [8], which consists of four subsets (HAdobe5k, HCOCO, HDay2night, and HFlickr) and includes 73146 samples for image harmonization. Each sample contains a composite image, a corresponding foreground mask, and a ground truth image. The HAdobe5k, HCOCO, HDay2night, and HFlickr subsets consist of 2160, 4283, 133, and 828 test images, respectively. These subsets have resolutions ranging from $312 \times 230$ to $6048 \times 4032$ pixels. The width and height of images in the HCOCO, HDay2night, and HFlickr subsets are all below 1024 pixels, and only the HAdobe5k subset is composed of images with a width or height larger than 1024 pixels.

We also train and evaluate our method on the ccHarmony dataset [25], which is constructed by transferring each image across different illumination conditions to simulate natural illumination variations. This dataset contains 3080 training samples and 1180 test samples, and each sample includes a composite image, a corresponding foreground mask, and a ground truth image.

Additionally, we evaluate our method against other methods on a real composite dataset [7], which consists of 100 test samples. Each sample contains only a composite image and a corresponding foreground mask. Note that all datasets used in our experiments are publicly available.

For the quantitative performance metrics, we calculate several key indicators, including Mean Square Error (MSE), foreground Mean Square Error (fMSE), Peak Signal-to-Noise Ratio (PSNR), and Structural Similarity Index (SSIM) for each individual image in the dataset. Subsequently, we compute the average values for the entire dataset and for each specific subset. While MSE serves as a crucial evaluation metric, it tends to be biased towards images that contain larger foreground regions due to the variations in foreground sizes across the dataset. This limitation makes MSE less reliable for a comprehensive assessment of image quality. In contrast, fMSE offers a more balanced and equitable evaluation of overall quality, making it a more suitable choice for our analysis [12].

Table 1: Quantitative comparison on the full-resolution test images of the iHarmony4 dataset. The best results are marked in bold, and the second best are underlined.

| Dataset | Metric | AdaInt [47] | SepLUT [48] | Harmonizer [20] | DCCF [46] | PCT-Net [12] | Our AICT |
|---|---|---|---|---|---|---|---|
| HAdobe5k | fMSE↓ | 216.12 | 208.11 | 196.12 | 195.54 | 149.39 | **138.45** |
| | MSE↓ | 30.38 | 26.53 | 24.37 | 23.12 | 19.35 | **17.09** |
| | PSNR↑ | 38.22 | 38.19 | 37.80 | 37.79 | 39.97 | **40.32** |
| | SSIM↑ | 0.9856 | 0.9844 | 0.9339 | 0.9858 | **0.9878** | **0.9878** |
| HCOCO | fMSE↓ | 374.89 | 380.77 | 374.96 | 317.43 | 245.67 | **240.62** |
| | MSE↓ | 21.76 | 22.43 | 20.93 | 16.85 | 12.45 | **12.30** |
| | PSNR↑ | 37.94 | 37.83 | 37.69 | 38.71 | **39.85** | 39.83 |
| | SSIM↑ | 0.9918 | 0.9912 | 0.9858 | 0.9930 | **0.9938** | 0.9936 |
| HDay2night | fMSE↓ | 699.88 | 702.16 | 640.74 | 715.43 | 700.65 | **639.00** |
| | MSE↓ | 49.40 | 52.98 | **37.28** | 55.78 | 46.47 | 42.90 |
| | PSNR↑ | 37.42 | 37.13 | 37.15 | **37.52** | 37.25 | 37.50 |
| | SSIM↑ | 0.9800 | 0.9782 | 0.9548 | 0.9788 | 0.9818 | **0.9819** |
| HFlickr | fMSE↓ | 588.40 | 580.50 | 479.26 | 438.49 | 357.53 | **334.11** |
| | MSE↓ | 86.08 | 86.69 | 69.19 | 64.63 | 45.79 | **43.74** |
| | PSNR↑ | 32.63 | 32.56 | 33.37 | 33.61 | 34.87 | **35.09** |
| | SSIM↑ | 0.9817 | 0.9790 | 0.9714 | 0.9844 | 0.9876 | **0.9877** |
| All | fMSE↓ | 358.29 | 358.51 | 339.23 | 302.56 | 238.27 | **228.43** |
| | MSE↓ | 31.96 | 31.36 | 27.62 | 24.72 | 18.80 | **17.76** |
| | PSNR↑ | 37.42 | 37.34 | 37.23 | 37.85 | 39.28 | **39.40** |
| | SSIM↑ | 0.9887 | 0.9876 | 0.9685 | 0.9897 | **0.9911** | 0.9910 |

### 4.1.2 Implementation Details

For the iHarmony4 dataset [8], our network is trained from scratch by using Adam optimizer with $\beta_1 = 0.9$, $\beta_2 = 0.999$, and $\epsilon = e^{-8}$. The batch size is set to 4 and the model is trained for 100 epochs. We set the learning rate as $5e^{-5}$ for the first 50 epochs and linearly decay it to zero over the next 50 epochs. For the network design, we set $K = 8$, $Q = 24$, $H' = 256$, and $W' = 256$. For the loss function, $\lambda$ is set to 0.01, and $A_{\min}$ is set to 1000 and 100 in the HR branch and LR branch, respectively. We resize training images to ensure that the length of their

Table 2: Quantitative comparison on the HAdobe5k subset at a $2048 \times 2048$ resolution. The best results are marked in bold, and the second best are underlined.

| Method | fMSE↓ | MSE↓ | PSNR↑ | SSIM↑ |
|---|---|---|---|---|
| iS$^2$AM [33] | 271.59 | 46.37 | 36.57 | 0.9838 |
| AdaInt [47] | 221.90 | 31.09 | 38.10 | 0.9847 |
| SepLUT [48] | 216.15 | 27.03 | 37.98 | 0.9834 |
| CDTNet-512 [7] | 159.13 | 23.35 | 38.45 | 0.9853 |
| Harmonizer [20] | 208.93 | 27.79 | 36.58 | 0.8992 |
| DCCF [46] | 197.23 | 23.00 | 38.34 | 0.9851 |
| PCT-Net [12] | 156.56 | 20.38 | 39.83 | 0.9870 |
| Our AICT | **147.99** | **17.92** | **40.07** | **0.9871** |

sides does not exceed 2048 pixels due to memory limitations. During the testing phase, we perform image harmonization using full-resolution images. Following [7], we also conduct tests on the HAdobe5k subset at two different resolutions ($1024 \times 1024$ and $2048 \times 2048$). For the ccHarmony dataset [25], we use the parameters trained on the iHarmony4 dataset as the initial parameters, and then fine-tune our network on the training set of the ccHarmony dataset. Following [25], the test image size in this dataset is set as $256 \times 256$. To augment training samples, we crop the composite image according to a random bounding box, the width and height of which are not smaller than the halved width and height of the composite image, respectively. The random horizontal flip is also applied to training samples. Our network is implemented based on the PyTorch framework and trained over approximately 75 hours on a computer equipped with two EPYC 7513 CPUs, 256GB of memory, and two 3090 GPUs.

### 4.2 Comparison with the State-of-the-arts

We conduct experiments and comparisons on the full-resolution test images of the iHarmony4 dataset [8]. AdaInt [47] and SepLUT [48], originally designed for image enhancement, are modified

and trained from scratch on the iHarmony4 dataset for the image harmonization task. Our AICT has three key differences compared to these methods. Firstly, they predict several weight parameters to fuse pre-trained LUTs, while our method dynamically predicts entire LUTs based on input. Secondly, they focus on global RGB-to-RGB transformations, whereas our AICT enables pixel-wise color transformations. Lastly, instead of applying adaptive interval learning and separable lookup tables for global sampling adjustments, our AICT achieves pixel-wise sampling interval adjustment to model local non-linearities in color transformation. To the best of our knowledge, only Harmonizer [20], DCCF [46], and PCT-Net [12] perform image harmonization on full-resolution images. Since Xu et al. [46] do not provide fMSE and SSIM values, we evaluate DCCF by running the provided code for comparison. Guerreiro et al. [12] propose two types of architectures: a CNN-based encoder-decoder network and a network based on a Visual Transformer (ViT) [10]. We choose the ViT-based network for comparison as it demonstrates better performance. The quantitative comparison results are shown in Table 1. We observe that AICT outperforms other methods on most metrics. Due to the limited amount of training samples in the HDay2night subset, our method achieves lower performance on this subset. To evaluate performance on low-resolution images, we also report quantitative results on the iHarmony4 dataset at a $256 \times 256$ resolution (see the Appendix). Additionally, we evaluate the harmonization performance on the HAdobe5k subset at a $2048 \times 2048$ resolution. As shown in Table 2, our method achieves the best performance across all metrics. Furthermore, we compare our method with others on this subset at a $1024 \times 1024$ resolution (see the Appendix).

As shown in Figure 6, we present the qualitative results of AICT against state-of-the-art methods on the full-resolution test images of the iHarmony4 dataset [8]. Compared to other methods, the images generated by AICT are closer to the ground truth images, making the composite images more realistic. More qualitative results are presented in the Appendix. Additionally, we also present the quantitative comparison results on the real composite dataset [25] to demonstrate the superiority and generalizability of our method (see the Appendix).

To further demonstrate the effectiveness of our method, we also present quantitative comparison results on the ccHarmony dataset [25] at $256 \times 256$ resolution. As shown in Table 3, our method also achieves the best performance across all metrics.

### 4.3 Ablation Study

We study the influence of the adaptive interval learning method on image harmonization performance using the iHarmony4 dataset [8], as shown in Table 4. We first remove the weight learning module and the adaptive interval learning method in AICT, denoted as "Single". Compared to AICT, this method achieves lower performance. We then incorporate the channel-crossing strategy [34, 35] into "Single", denoted as "Cross". The parameter model uses an RGB image to predict parameter maps, which inherently include information from all color channels. Thus, the channel-crossing strategy introduces redundancy without improving performance. We also remove the adap-

Table 3: Quantitative comparison on the ccHarmony dataset at a $256 \times 256$ resolution. The best results are marked in bold, and the second best are underlined.

| Method | fMSE↓ | MSE↓ | PSNR↑ |
|---|---|---|---|
| DoveNet [8] | 880.94 | 110.84 | 31.64 |
| RainNet [22] | 519.32 | 58.11 | 34.78 |
| IIH [15] | 636.28 | 83.72 | 33.64 |
| D-HT [14] | 514.47 | 55.73 | 35.07 |
| iS$^2$AM [33] | 264.84 | 28.83 | 36.05 |
| CDTNet [7] | 264.51 | 27.87 | 36.62 |
| Harmonizer [20] | 402.09 | 43.31 | 34.68 |
| DCCF [46] | 259.83 | 29.25 | 36.62 |
| GiftNet [25] | 235.20 | 24.55 | 37.59 |
| Our AICT | **232.66** | **24.14** | **38.46** |

tive interval learning method in AICT, denoted as "w/o Int". Compared to AICT, the performance of this method decreases due to the reduced expressiveness of the curves. To investigate whether using more LUTs can improve performance, we first cascade 2 and 3 LUTs for adaptive interval learning, denoted as "Int × 2" and "Int × 3", respectively. Then, we cascade 2 and 3 LUTs for color transformation, denoted as "Tra × 2" and "Tra × 3", respectively. Finally, we cascade 4 and 6 LUTs for alternating adaptive interval learning and color transformation, denoted as "Alt × 2" and "Alt × 3", respectively. The experimental results show that increasing the number of LUTs for adaptive interval learning does not improve performance. Furthermore, using more LUTs for color transformation decreases performance due to increased training difficulty. Additionally, we combine AdaInt [47] and SepLUT [48] for global RGB-to-RGB transformations, denoted as "AdaInt

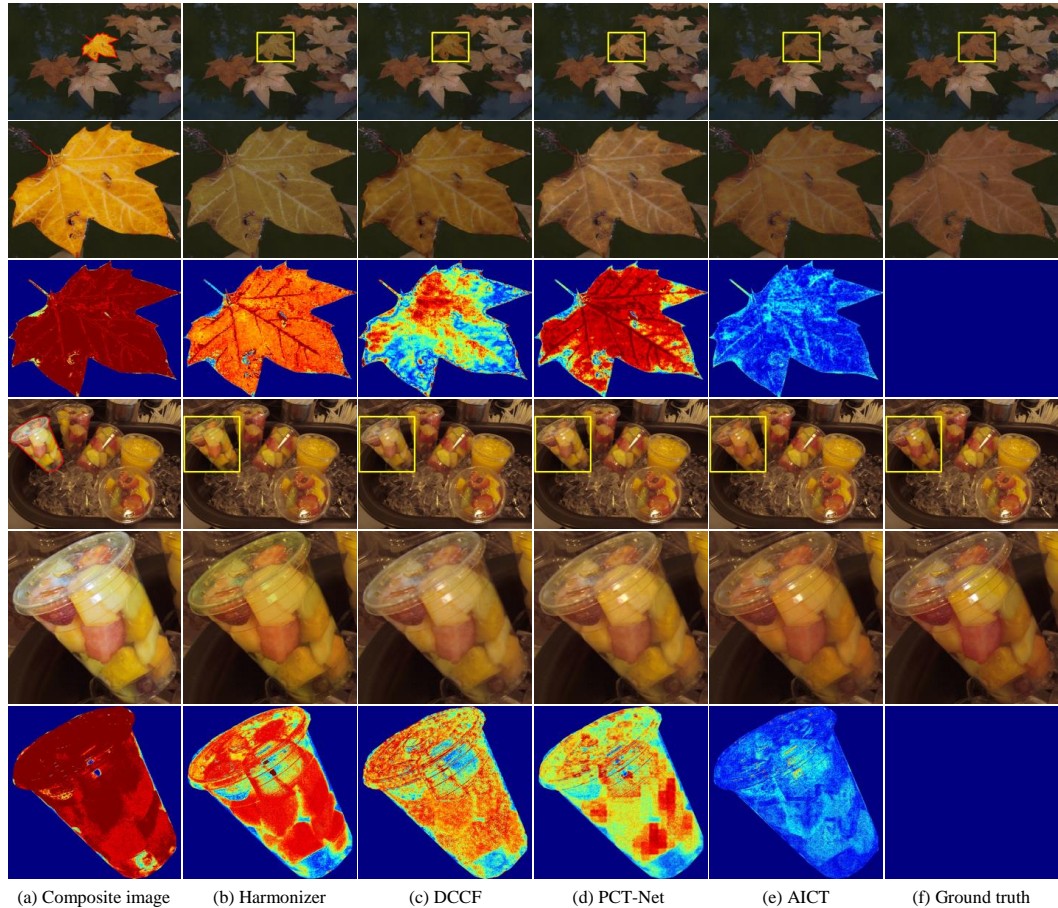

| (a) Composite image | (b) Harmonizer | (c) DCCF | (d) PCT-Net | (e) AICT | (f) Ground truth |

Figure 6: Qualitative comparison results and error maps. We visualize the error between the harmonized images and the ground truth images. The error maps are normalized for display, and the foreground is outlined in red.

+ SepLUT". Compared to AICT, one reason for the performance decline of "AdaInt + SepLUT" is the neglect of local context.

We conduct ablation studies on the global consistent weight learning method, as shown in Table 5. First, we remove the weight vector corresponding to the R channel, denoted as "w/o R". Next, we remove the weight vectors for both the R and G channels, denoted as "w/o RG". Finally, we remove the entire weight learning module, denoted as "w/o Weight". As the number of removed weight vectors increases, the harmonization performance decreases, highlighting the importance of the global consistent weight learning method. We also predict spatially varying weight parameters to weight each sampling point of the parameter map $R$, denoted as "Spatial". The results for "Spatial" indicate that learning spatially varying weight parameters does not improve performance, as the LUT is responsible for pixel-wise color transformations. These observations demonstrate that the proposed adaptive interval learning method and global consistent weight learning method are effective.

Table 4: Ablation studies on the adaptive interval learning method. The best results are marked in bold.

| Method | fMSE↓ | MSE↓ | PSNR↑ |
|---|---|---|---|
| Single | 248.51 | 19.71 | 39.15 |
| Cross | 264.33 | 21.40 | 38.90 |
| w/o Int | 239.34 | 18.96 | 39.20 |
| Int × 2 | 241.79 | 19.24 | 39.17 |
| Int × 3 | 231.91 | 18.04 | 39.38 |
| Tra × 2 | 456.98 | 33.94 | 36.16 |
| Tra × 3 | 487.60 | 36.03 | 35.91 |
| Alt × 2 | 441.27 | 33.09 | 36.28 |
| Alt × 3 | 489.02 | 35.95 | 35.94 |
| AdaInt + SepLUT | 357.61 | 31.11 | 37.47 |
| Our AICT | **228.43** | **17.76** | **39.40** |

## 4.4 Hyper-parameter Analyses

We conduct studies on key parameters using the iHarmony4 dataset [8], including the number of knots $K$, the coefficient of the LR branch loss $\lambda$, and the hyper-parameter $A_{\min}$ in the foreground-normalized MSE loss, as shown in Table 6. We set $K$ to 6, 8, and 10 to analyze the influence of curves with different numbers of knots for the image harmonization task, where $K$ is set to 8 in AICT. By comparing "K=6" with AICT, we observe that increasing the number of knots reduces fMSE and

Table 5: Ablation studies on the global consistent weight learning method. The best results are marked in bold.

| Method | fMSE↓ | MSE↓ | PSNR↑ |
|---|---|---|---|
| w/o R | 230.83 | 18.25 | 39.39 |
| w/o RG | 232.20 | 19.10 | 39.18 |
| w/o Weight | 247.48 | 19.40 | 39.16 |
| Spatial | 230.24 | 18.03 | 39.23 |
| Our AICT | **228.43** | **17.76** | **39.40** |

MSE while improving PSNR, indicating that a larger number of knots enhances the harmonization ability of the curves. However, as the number of knots increases, the performance of "K=10" decreases, suggesting that an excessive number of knots increases network parameters. This complicates the training process of the network, making it more challenging to achieve optimal performance. To investigate the effect of the LR branch loss, we set $\lambda$ to 0, 0.01, and 0.1, where $\lambda$ set to 0.01 in AICT. When $\lambda$ is set to 0.01, AICT achieves the best performance on all metrics, demonstrating that an optimal weight for the LR branch loss is crucial for the high-resolution image harmonization. For the foreground-normalized MSE loss, we set $A_{\min}$ to $H \times W$ and $H' \times W'$ in the HR and LR branch, respectively, which means these objective functions are equivalent to MSE functions, denoted as "MSE". The results indicate that using the foreground-normalized MSE loss improves harmonization performance, as it prevents training samples with foregrounds of different sizes from being trained with varying loss magnitudes, ensuring effective training for small foreground images.

## 5 Conclusion

In this paper, we formulate image harmonization as an image-based multiple curve estimation problem and propose an Adaptive-Interval Color Transformation method, which predicts pixel-wise color transformation and adaptively adjusts the sampling interval to model local non-linearities of the color transformation at high resolution. Specifically, a parameter network is first designed to generate multiple curves as position-dependent 3D LUTs, which use the color and position of each pixel to perform pixel-wise color transformation. Then, we separate a color transform into a cascade of sub-transformations

Table 6: Hyper-parameter analyses on the number of knots $K$, the coefficient of the LR branch loss $\lambda$, and the hyper-parameter $A_{\min}$ in the foreground-normalized MSE loss. The best results are marked in bold.

| Method | fMSE↓ | MSE↓ | PSNR↑ |
|---|---|---|---|
| $K = 6$ | 241.67 | 18.64 | 39.20 |
| $K = 10$ | 245.43 | 19.69 | 39.15 |
| $\lambda = 0$ | 272.56 | 21.24 | 38.70 |
| $\lambda = 0.1$ | 253.34 | 20.39 | 39.02 |
| MSE | 287.04 | 23.17 | 38.41 |
| Our AICT | **228.43** | **17.76** | **39.40** |

using two position-dependent 3D LUTs to achieve the non-uniform sampling intervals of the color transform. Finally, a global consistent weight learning method is proposed to predict an image-level weight for each color transform, utilizing global information to enhance the overall harmony. Experimental results demonstrate that our method achieves state-of-the-art performance in high-resolution image harmonization with a lightweight architecture.

## Acknowledgements

This work was supported in part by the National Natural Science Foundation of China under Grants 62272134, 62236003 and 62072141, in part by the National Science and Technology Major Project under Grant 2021ZD0110901, in part by Shenzhen Colleges and Universities Stable Support Program under Grant GXWD20220817144428005, and in part by the Natural Science Foundation of Shandong Province under Grant ZR2024QF065.

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

# A Appendix

## A.1 More Quantitative Results

Table 7: Quantitative comparison on the iHarmony4 dataset at a $256 \times 256$ resolution. The best results are marked in bold, and the second best are underlined.

| Method | HAdobe5k | | | HCOCO | | | HDay2night | | | HFlickr | | | All | | |
|---|---|---|---|---|---|---|---|---|---|---|---|---|---|---|---|
| | fMSE↓ | MSE↓ | PSNR↑ | fMSE↓ | MSE↓ | PSNR↑ | fMSE↓ | MSE↓ | PSNR↑ | fMSE↓ | MSE↓ | PSNR↑ | fMSE↓ | MSE↓ | PSNR↑ |
| DoveNet [8] | 380.39 | 52.32 | 34.34 | 551.01 | 36.72 | 35.83 | 1075.71 | 54.05 | 35.18 | 827.03 | 133.14 | 30.21 | 549.96 | 52.36 | 34.75 |
| BargainNet [6] | 279.66 | 39.94 | 35.34 | 397.85 | 24.84 | 37.03 | 835.63 | 50.98 | 35.67 | 698.40 | 97.32 | 31.34 | 405.23 | 37.82 | 35.88 |
| IIH [15] | 284.21 | 43.02 | 35.20 | 416.38 | 24.92 | 37.16 | 797.04 | 55.53 | 35.96 | 716.60 | 105.13 | 31.34 | 400.29 | 38.71 | 35.90 |
| D-HT [14] | 265.11 | 38.53 | 36.88 | 299.30 | 16.89 | 38.76 | 704.42 | 53.01 | 37.10 | 515.45 | 74.51 | 33.13 | 320.78 | 30.30 | 37.55 |
| D-HT+ [13] | 242.56 | 36.83 | 37.17 | 274.66 | 37.10 | 36.83 | 736.58 | 49.68 | 36.68 | 471.06 | 67.88 | 33.55 | 295.56 | 27.89 | 37.94 |
| iS$^2$AM [33] | 173.96 | 21.88 | 38.08 | 266.19 | 16.48 | 39.16 | 590.97 | 40.59 | 37.72 | 443.65 | 69.67 | 33.56 | 264.96 | 24.44 | 38.19 |
| iDIH+HRNet [33] | N/A | 21.36 | 37.35 | N/A | 14.01 | 39.64 | N/A | 50.61 | 37.68 | N/A | 60.41 | 34.03 | 252.00 | 22.00 | 38.31 |
| CDTNet [7] | N/A | 20.62 | 38.24 | N/A | 16.25 | 39.15 | N/A | 36.72 | 37.95 | N/A | 68.61 | 33.55 | 252.05 | 23.75 | 38.23 |
| S2CRNet [21] | N/A | 34.91 | 36.42 | N/A | 23.22 | 38.48 | N/A | 51.67 | 36.81 | N/A | 98.73 | 32.38 | N/A | 35.58 | 37.18 |
| Harmonizer [20] | 170.05 | 21.89 | 37.64 | 298.42 | 17.34 | 38.77 | 542.07 | 33.14 | 37.56 | 434.06 | 64.81 | 33.63 | 280.51 | 24.26 | 37.84 |
| DCCF [46] | 172.49 | 23.43 | 37.18 | 272.10 | 14.87 | 39.52 | 655.46 | 45.09 | 38.08 | 411.56 | 61.42 | 33.84 | 265.52 | 22.05 | 38.50 |
| Sg-MHH [31] | N/A | 22.04 | 38.64 | N/A | 13.95 | 39.14 | N/A | 43.57 | 36.86 | N/A | 59.03 | 34.00 | N/A | 21.89 | 38.38 |
| SCS-Co [16] | 165.48 | 21.01 | 38.29 | 245.54 | 13.58 | 39.88 | 606.80 | 41.75 | 37.83 | 393.72 | 55.83 | 34.22 | 258.86 | 21.33 | 38.75 |
| PCT-Net [12] | 157.24 | 21.25 | 39.97 | 208.26 | 10.72 | 40.78 | 654.81 | 44.74 | 37.65 | 341.10 | 44.30 | 35.13 | 216.25 | 18.16 | 39.85 |
| GKNet [32] | 138.22 | 17.84 | 39.97 | 222.31 | 12.95 | 40.32 | 546.06 | 42.76 | 38.47 | 372.90 | 57.58 | 34.45 | 220.44 | 19.90 | 39.53 |
| GiftNet [25] | 143.96 | 18.35 | 38.76 | 229.68 | 12.70 | 39.91 | 566.47 | 38.28 | 37.81 | 360.08 | 54.33 | 34.44 | 225.30 | 19.46 | 38.92 |
| LCS [45] | 154.82 | 20.11 | 38.93 | 217.55 | 11.27 | 39.94 | 587.44 | 39.79 | 38.42 | 386.12 | 54.20 | 34.76 | 238.19 | 20.77 | 39.36 |
| Our AICT | 133.15 | 16.50 | 40.55 | 206.24 | 10.74 | 40.68 | 594.92 | 41.27 | 37.93 | 320.45 | 42.58 | 35.33 | 204.67 | 16.53 | 39.99 |

In this section, we provide supplementary quantitative results to further demonstrate the advantages of our proposed method. First, we evaluate AICT against more state-of-the-art methods using low-resolution test images from the iHarmony4 dataset [8]. Following previous work, we downsample these images to $256 \times 256$ pixels to obtain low-resolution versions. It is important to note that AICT employs parameters trained on high-resolution images from the iHarmony4 dataset [8] and uses low-resolution images as input to perform pixel-wise color transformation instead of downsampling high-resolution output images for comparison. As shown in Table 7, AICT achieves superior performance across most metrics, which demonstrates the effectiveness of our method on low-resolution images. However, our method performs slightly worse on the HDay2night subset due to the limited number of training images in this subset.

We also compare AICT with state-of-the-art methods on the HAdobe5k subset at a resolution of $1024 \times 1024$. As shown in Table 8, our method achieves the best performance in terms of MSE and PSNR, and performs slightly worse than CDTNet-256 [7] in terms of fMSE and SSIM. It is important to note that CDT-Net [7] requires retraining for images of varying resolutions. In contrast, AICT demonstrates flexibility by using parameters trained on high-resolution images from the iHarmony4 dataset [8] to effectively process images at various resolutions.

In order to compare AICT with state-of-the-art methods at ultra-high resolutions, we collect two ultra high-resolution benchmarks (over4K and over5K) by selecting images with res-

Table 8: Quantitative comparison on the HAdobe5k subset at a $1024 \times 1024$ resolution. The best results are marked in bold, and the second best are underlined.

| Method | fMSE↓ | MSE↓ | PSNR↑ | SSIM↑ |
|---|---|---|---|---|
| pix2pixHD [42] | 332.43 | 63.45 | 31.64 | 0.9135 |
| CRD [4] | 259.28 | 90.11 | 29.77 | 0.8225 |
| HiDT [1] | 1501.93 | 265.32 | 29.95 | 0.9628 |
| DoveNet [8] | 312.88 | 51.00 | 34.81 | 0.9729 |
| S$^2$AM [9] | 262.39 | 47.01 | 35.68 | 0.9784 |
| IIH [15] | 417.33 | 56.34 | 34.69 | 0.9471 |
| RainNet [22] | 305.17 | 42.56 | 36.61 | 0.9844 |
| iS$^2$AM [33] | 168.85 | 25.03 | 38.29 | 0.9846 |
| CDTNet-256 [7] | 152.13 | 21.24 | 38.77 | 0.9868 |
| AdaInt [47] | 235.86 | 31.75 | 37.81 | 0.9837 |
| SepLUT [48] | 224.99 | 27.41 | 37.75 | 0.9826 |
| Harmonizer [20] | 229.17 | 32.67 | 35.75 | 0.9081 |
| DCCF [46] | 196.41 | 22.63 | 38.39 | 0.9846 |
| PCT-Net [12] | 171.04 | 22.57 | 39.39 | 0.9860 |
| Our AICT | 156.81 | 19.50 | 39.67 | 0.9864 |

olutions exceeding 4K ($4096 \times 2160$) and 5K ($5120 \times 2880$) from the HAdobe5k subset. As shown in Table 9, our method achieves the best performance on the over4K benchmark in terms of fMSE, MSE, and SSIM, and the second-best performance in terms of SSIM. As shown in Table 10, on the over5K benchmark, our method achieves the best performance in terms of MSE and PSNR and the second-best performance in terms of fMSE and SSIM.

We further compare the FLOPs, memory cost, and inference time of our method with state-of-the-art methods on the HAdobe5k subset. These metrics are crucial for evaluating the computational efficiency and scalability of the models. Our tests were conducted on a computer equipped with 32GB of memory and a GeForce GTX 1070 Ti GPU. Tables 11 and 12 show the comparison results at $1024 \times 1024$ and $2048 \times 2048$ resolutions, respectively.

Table 9: Quantitative comparison on the HAdobe5k subset at resolutions exceeding 4K ($4096 \times 2160$). The best results are marked in bold, and the second best are underlined.

| Method | fMSE↓ | MSE↓ | PSNR↑ | SSIM↑ |
|---|---|---|---|---|
| AdaInt [47] | 226.67 | 27.84 | 38.20 | 0.9882 |
| SepLUT [48] | 219.61 | 24.98 | 38.13 | 0.9866 |
| Harmonizer [20] | 201.44 | 23.06 | 37.89 | 0.9368 |
| DCCF [46] | 199.85 | 20.55 | 38.39 | 0.9885 |
| PCT-Net [12] | 165.31 | 19.36 | 39.83 | **0.9900** |
| Our AICT | **154.45** | **16.79** | **40.16** | 0.9898 |

At both resolutions, Harmonizer [20] has a significant advantage in terms of FLOPs, while DCCF [46] has the lowest memory cost. At a resolution of $1024 \times 1024$, our method is comparable to PCT-Net [12] in FLOPs, memory cost, and inference time. However, at a resolution of $2048 \times 2048$, our method demonstrates lower memory cost and shorter inference time compared to PCT-Net. One important reason is that the parameter maps in our method have a fixed size, whereas PCT-Net needs to align the parameter maps with the composite images, resulting in higher memory consumption and longer inference time.

## A.2   More Qualitative Results

We present additional qualitative comparison results on the iHarmony4 dataset [8], shown in Figure 7. Specifically, we compare our proposed method with Harmonizer [20], DCCF [46], and PCT-Net [12]. The qualitative results demonstrate that our method achieves superior harmonization performance, generating images that closely resemble the ground truth. Our method excels in adjusting the overall brightness and preserving local details in the foreground, leading to more realistic and visually pleasing composite images.

Table 10: Quantitative comparison on the HAdobe5k subset at resolutions exceeding 5K ($5120 \times 2880$). The best results are marked in bold, and the second best are underlined.

| Method | fMSE↓ | MSE↓ | PSNR↑ | SSIM↑ |
|---|---|---|---|---|
| AdaInt [47] | 330.25 | 29.26 | 36.35 | 0.9782 |
| SepLUT [48] | 395.42 | 22.06 | 36.30 | 0.9775 |
| Harmonizer [20] | 312.87 | 36.78 | 36.55 | 0.9437 |
| DCCF [46] | 320.03 | 28.44 | 35.73 | 0.9757 |
| PCT-Net [12] | **229.63** | 21.41 | 37.49 | **0.9790** |
| Our AICT | 293.09 | **15.49** | **37.64** | 0.9786 |

Table 11: Comparison of inference time, memory cost, and FLOPs at a $1024 \times 1024$ resolution.

| Image Size | Method | FLOPs↓ (G) | Memory↓ (MB) | Inference Time↓ (ms) |
|---|---|---|---|---|
| 1024 × 1024 | Harmonizer [20] | 0.036 | 1034 | 15.04 |
| | DCCF [46] | 12.68 | 889 | 37.66 |
| | PCT-Net [12] | 13.04 | 1280 | 72.01 |
| | Our AICT | 14.35 | 1289 | 72.32 |

Table 12: Comparison of inference time, memory cost, and FLOPs at a $2048 \times 2048$ resolution.

| Image Size | Method | FLOPs↓ (G) | Memory↓ (MB) | Inference Time↓ (ms) |
|---|---|---|---|---|
| 2048 × 2048 | Harmonizer [20] | 0.036 | 1959 | 54.68 |
| | DCCF [46] | 12.68 | 957 | 36.94 |
| | PCT-Net [12] | 13.04 | 2078 | 99.08 |
| | Our AICT | 14.35 | 1653 | 91.64 |

## A.3   Evaluation on Real Composite Images

We provide qualitative results on a real composite dataset [7], which presents a significant challenge by combining foreground regions from different light fields with background regions. The resolution of this dataset ranges from $1024 \times 1024$ to $6016 \times 4000$ pixels. As shown in Figure 8, AICT outperforms other methods in producing visually superior results in real-world scenarios. The images generated by AICT show a more harmonious appearance, with both overall brightness and local foreground

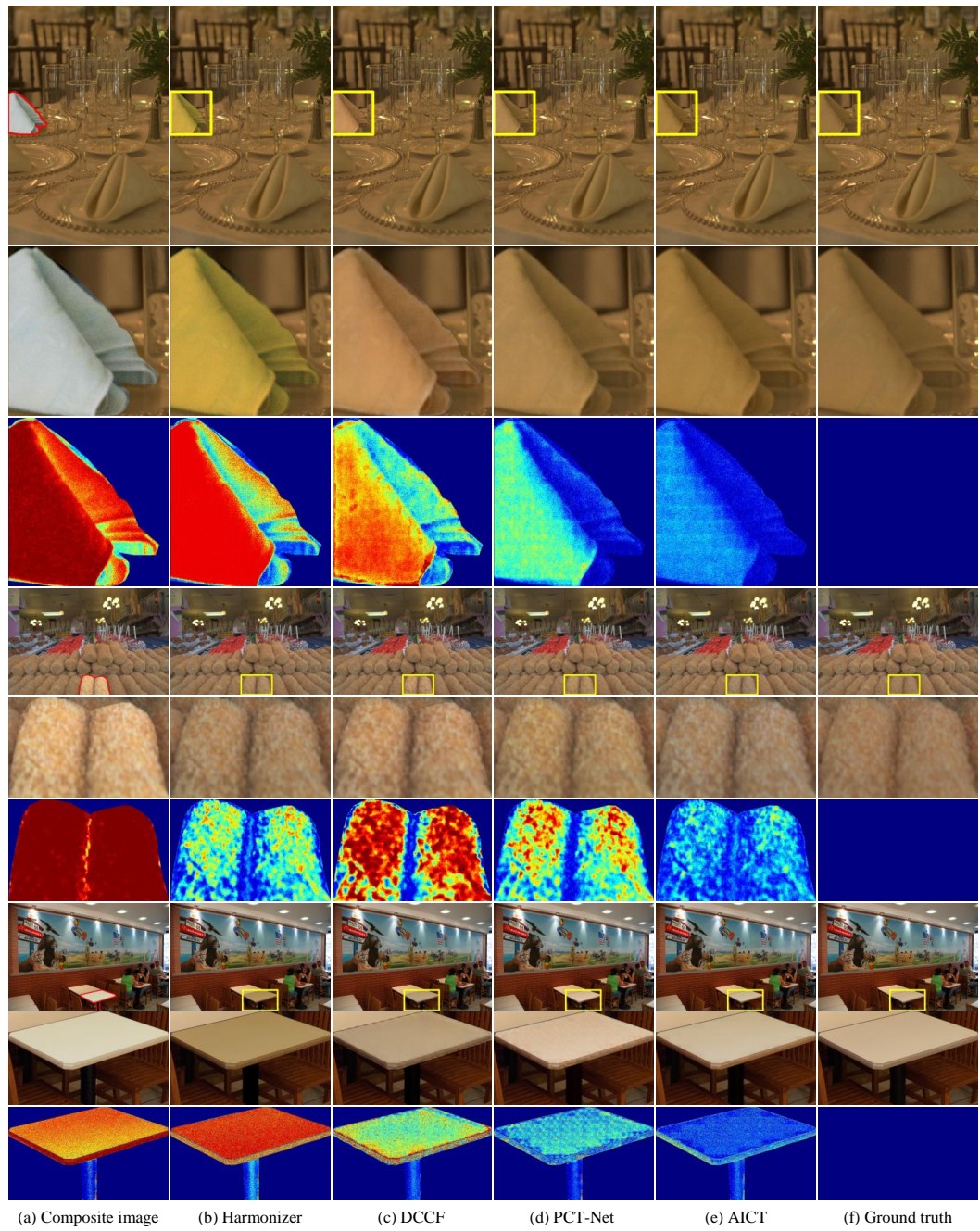

| (a) Composite image | (b) Harmonizer | (c) DCCF | (d) PCT-Net | (e) AICT | (f) Ground truth |

Figure 7: Qualitative comparison results and error maps. We visualize the error between the harmonized images and the ground truth images. The error maps are normalized for display, and the foreground is outlined in red.

Table 13: User study results. The best result is marked in bold.

| Method | Harmonizer [20] | DCCF [46] | PCT-Net [12] | Our AICT |
|--------|-----------------|-----------|--------------|----------|
| Score↑ | 21.60 | 28.00 | 32.40 | **37.80** |

details appearing more realistic. Following [39], we first randomly select 20 high-resolution real composite images from this dataset to conduct a user study. Then, 20 volunteers independently rank

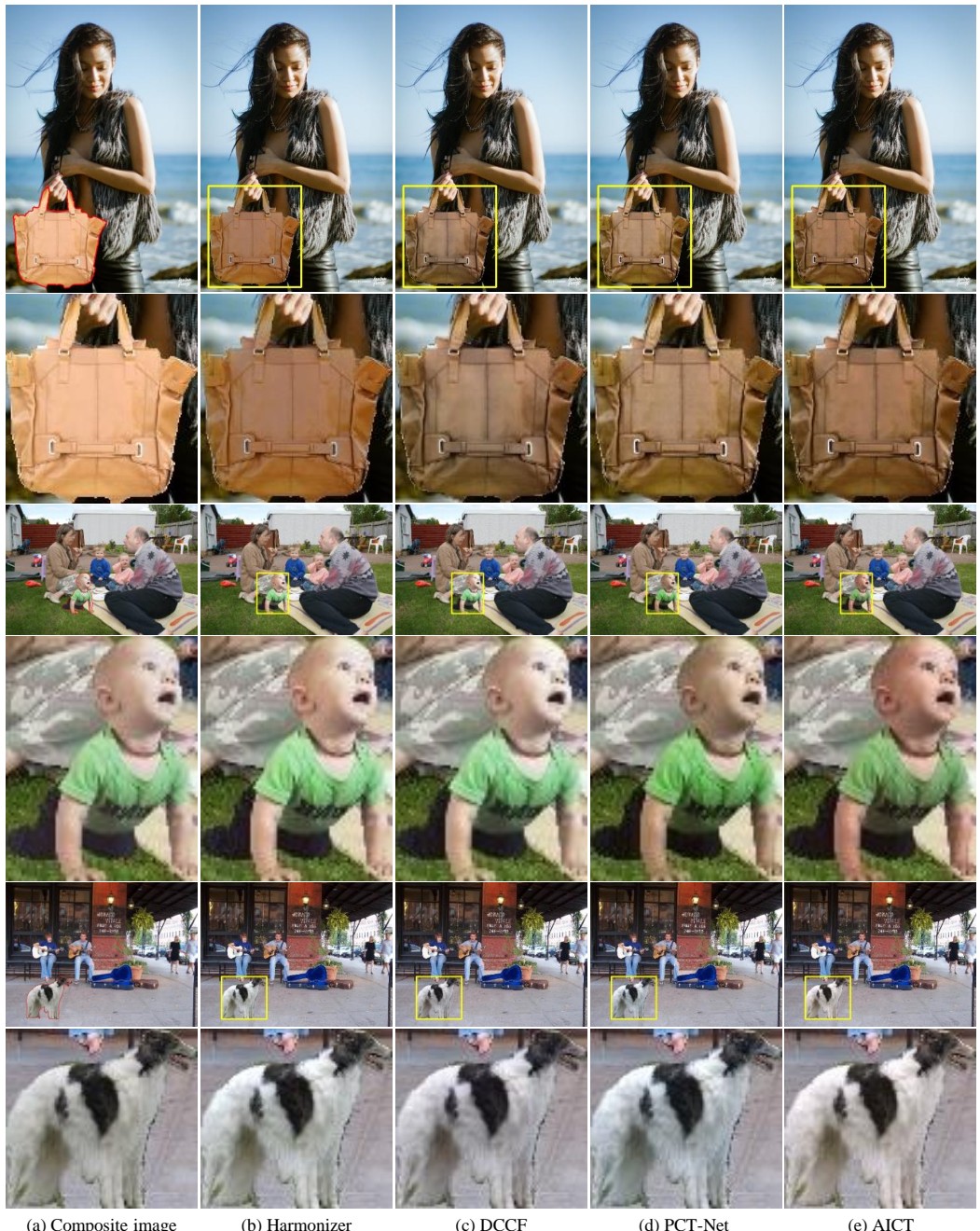

| (a) Composite image | (b) Harmonizer | (c) DCCF | (d) PCT-Net | (e) AICT |

Figure 8: Qualitative comparison results against state-of-the-art methods on the real composite dataset. The foreground is outlined in red.

the predictions from 1 to 3 based on visual quality. Scores of 3, 2, and 1 are assigned for ranks 1, 2, and 3, respectively. The mean scores for each method are presented in the table below. As shown in Table 13, our method achieves the highest score. The above experimental results demonstrate the superiority and generalizability of our method.

## A.4   Broader Impacts

This paper proposes a pixel-wise high-resolution image harmonization method aimed at adjusting the color of the foreground to seamlessly integrate with the background, thereby enhancing the realism

of composite images and making a significant contribution to the image composition community. Simultaneously, our research holds paramount importance in domains such as art, entertainment, and commerce. However, it is worth noting that image harmonization techniques could potentially be exploited to create deceptive or misleading visual content. To address this concern, there is a pressing need for the development of image manipulation detection methods. Additionally, we emphasize the importance of raising awareness about the capabilities and limitations of such methods to effectively mitigate their adverse impacts.

## A.5 Limitations

A primary limitation of our method lies in its incapability to effectively address foreground elements like mirrors and glass. This is because the model cannot accurately simulate the interplay between light and various materials in such scenarios. Figure 9 illustrates some failure cases where our method struggles to harmonize images containing such elements.

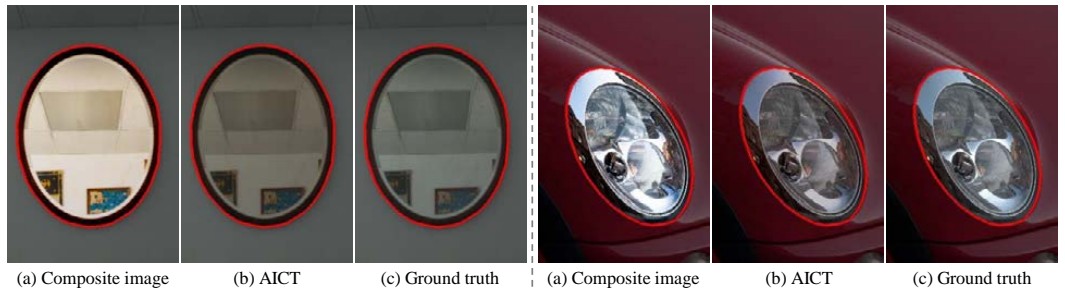

| (a) Composite image | (b) AICT | (c) Ground truth | (a) Composite image | (b) AICT | (c) Ground truth |

Figure 9: Examples of failed harmonization on the iHarmony4 dataset.

