# OpenReview forum: "High-Resolution Image Harmonization with Adaptive-Interval Color Transformation"
_NeurIPS.cc/2024/Conference — NeurIPS 2024 poster_

### Official Review · Reviewer_M5vB · 2024-06-13

**Soundness:** 3
**Presentation:** 3
**Contribution:** 3
**Rating:** 5
**Confidence:** 5

**Summary:**

This work proposes the Adaptive-Interval Color Transformation (AICT) method for harmonizing high-resolution composite images. The proposed AICT first uses a parameter network to predict multiple curves (as 3D LUTs) to perform pixel-wise color transformation at low-resolution. The AICT then adjusts the sampling intervals of the color transformation by splitting each predicted 3D LUT into two 3D LUTs. Last, AICT uses a global consistent weight learning method to predict an image-level weight for each color transform. Experiments on the iHarmony4 [8] and ccHarmony [26] datasets show promising quantitative results of the proposed AICT.

**Strengths:**

Originality:
(+) The strength is that the difference to existing HR image harmonization methods is clear.

Quality:
(+) The proposed method is complete and technically sound. (+) The experiment settings (evaluation datasets, metrics, and competing methods) are convincing.

Clarity:
(+) The paper presents sufficient implementation details for reproducing the proposed method.

Significance:
(+) Quantitative results in Tables 1-3, and 6-7 show superior or comparable performance of AICT in harmonizing HR and LR composite images.

**Weaknesses:**

Originality:
This work proposes the AICT method to address an existing task of harmonizing high-resolution (HR) composite images. While previous HR image harmonization methods (e.g., [12,46]) predict and then upsample low-resolution (LR) parameter maps for harmonization, the proposed AICT also predicts LR parameter maps but applies two 3D LUTs to harmonize the colors in a coarse-to-fine manner. (-) One weakness is the limited/vague novelty of using two 3D LUTs for color transformation, which is a combination of ideas from existing LUT-based image enhancement methods [47, 48] but lacks necessary discussions and experimental verifications.

Quality:
(-) The paper mentions the ``non-linearities of the color transform at high resolution’’ many times as the key problem suffered by existing methods and addressed by the proposed AICT. However, the paper does not explain what the ``non-linearities’’ are, nor do the visual results in Figures 7, 8, and 9 illustrate such advantages.
(-) The paper claims AICT being lightweight and computationally efficient but Figure 2 only demonstrates that AICT has a small model size. According to Tables 8 and 9, AICT runs much slower than [21] and [46] and is comparable to [12]. Hence, such a claim needs to be tuned down.
(-) The paper claims the ``global consistent weight method’’ to be novel but the paper does not explain its novelty.

Clarity:
There are a few weak points regarding the paper writing.
(-) The paper mentions the ``adaptive interval learning method’’ and the ``global consistent weight method’’ as the second main contribution (line 86-87, page 2). However, the Method section discusses the proposed ``global consistent weight method’’ very briefly as a part of subsection 3.2 (line 207-214, page 5). The authors are suggested to either explain the ``global consistent weight method’’ in detail regarding its novelty or remove the corresponding claim in the second contribution.
(-) According to Figure 3 and Section 3 (line 134-144, page 4) the LR branch of the proposed AICT outputs two parameter maps (C and F). However, Eq. (3) indicates that there is a low-resolution harmonized image predicted by the LR branch.
(-) Figure 1 is not self-contained. The S2CRNet [22] is missing in Figure 2.
(-) The sub-title of Section 2.1 is better: ``High-resolution Image Harmonization.
(-) There are no discussions on the relations between the proposed AICT and existing methods in Section 2.1 and 2.2.
(-) There are many repeated sentences in, for example,(Line 5-6, Line 60-63, Line 374-378), and (Line 7-13, Line 67-78).
(-) The references contain redundant items, such as [13,14].
(-) ``M=6’’ and ``M=10’’ should be ``K=6’’ and ``K=10’’.
(-) It is better to move the Tables 6 and 7 to the main paper.

Significance:
(-) The visual results in Figure 7-9 do not show significant improvements.
(-) The main quantitative performance gain seems to come from the foreground-normalized MSE loss [34], according to Tables 4 and 5.
The paper presents a harmonization method that can produce better quantitative results, but the visual advantages illustrated in this paper are not significant

**Questions:**

Q1: What is the novelty of the proposed 3D LUTs-based adaptive-interval color transformation, compared to the non-uniform sampling interval learning in Yang et al. [47] and the decomposition of a single color transform into two sub-transformations in Yang et al. [48]. It is suggested to explain the challenges of directly combining [47] and [48], and how the proposed AICT addresses such challenges. Comparison results between AICT and a simple combination of [47] and [48] are suggested to provide.
Q2: Can authors report and explain the results of incorporating more 3D LUTs?
Q3: According to Tables 4 and 5, the foreground-normalized MSE loss [34] is more important to the final results compared to the ``w/o Weight’’ and ``w/o Color’’. Can authors explain this and/or provide more results for clarification?
Q4: Any more ablations on the proposed ``global weight’’?
Q5: Can authors report non-reference image quality metrics (e.g., NIQE and LPIPS) for the harmonized real composite images in Figure 9?
Q6: Can authors provide intermediate visual results of the ablated models?
Q7: Can authors explain why using the channel-crossing strategy does not improve the performance?
Q8: Any reason not to use SSIM as the evaluation metric?

**Limitations:**

The paper has discussed the limitations and potential negative societal impact. I cannot think of any further critical points.

---

> ### Author Rebuttal · Authors · 2024-08-06
>
> **Thanks for your positive evaluation and valuable suggestions.**
>
> ***W1 & Q1: Limited novelty of using two 3D LUTs for color transformation, which is a combination of ideas from existing LUT-based image enhancement methods [47, 48] but lacks necessary discussions and experimental verifications..***
> ***R1 & A1:*** Regarding the novelty of using two 3D LUTs for color transformation, we request the reviewer to refer to our response to Q2 & Q3 of Reviewer pwQP due to the space limitation.
>
> Our method is not a combination of  Adaint [47] and Seplut [48]. In fact, simply combining them faces the challenges of limited color transformation capability and overlooking diverse local contexts. Our AICT addresses these challenges by dynamically predicting entire LUTs for pixel-wise sampling interval adjustment and color transformations.
> We also conduct experimental comparison between our method and Adaint and Seplut as well as their combination. As shown in the table below, our AICT achieves the best performance in terms of all metrics.
>
> |Method|fMSE↓|MSE↓|PSNR↑|SSIM↑|
> |----|:----:|:----:|:----:|:----:|
> |Adaint|358.28|31.96|37.41|0.9886|
> |Seplut|358.51|31.35|37.33|0.9876|
> |Adaint+Seplut|252.32|30.55|37.66|0.9889|
> |our AICT|**228.43**|**17.76**|**39.40**|**0.9910**|
>
> ***W2: Explain “the non-linearities”  and show the visual results.***
> ***R2:*** Due to resolution differences,  predicted 3D LUTs cannot align with the original input image. Consequently, linear interpolation based on a few sampling points is performed within the 3D LUT to obtain output color values, leading to "non-linear" transformations. To illustrate this advantage, we present the color distribution in local areas of high-resolution harmonized and ground truth images in Figure 3 of the PDF attached in the author rebuttal, which shows that the pixel values predicted by our method are closer to the real color distribution.
>
> ***W3: “lightweight and computationally efficient” needs to be tuned down.***
> ***R3:*** Thanks for this suggestion. We will tune down our claim in the revised manuscript.
>
> ***W4 & Q4: Explain novelty of the 'global consistent weight method'.***
> ***R4 & A4:*** We request the reviewer to refer to our response to Q4 & Q5 & Q6 of the Reviewer pwQP due to space limitation.
>
> ***W5: There are a few weak points regarding the paper writing.***
> ***R5:*** Thanks for this suggestion. We will improve the paper writing in the revised manuscript.
>
> ***W6: The visual results do not show significant improvements.***
> ***R6:*** To better show significant improvements, we quantify and visualize the error between the harmonized and ground truth images as shown in Figure 1 of the PDF attached in the author rebuttal. We also present additional visual results in Figure 2 of the PDF.
>
> ***Q2: Report and explain the results of incorporating more 3D LUTs.***
> ***A2:*** We report more results in the table below. For each color channel, we use only 1 LUT for color transformation ('Single') and cascade 2 and 3 LUTs for adaptive interval learning ('Int x 2', 'Int x 3') and color transformation ('Tra x 2', 'Tra x 3'). We also cascade 4 and 6 LUTs for alternating adaptive interval learning and color transformation ('Alt x 2', 'Alt x 3'). Performance improves slightly with more LUTs but plateaus at three, due to increased training difficulty.
>
> |Method|fMSE↓|MSE↓|PSNR↑|SSIM↑|
> |----|:----:|:----:|:----:|:----:|
> |Single |248.51|19.71|39.15|0.9908|
> |Int x 2|227.95|**17.40**|**39.50**|**0.9910**|
> |Int x 3|229.50|18.55|38.95|0.9909|
> |Tra x 2|**228.40**|*17.50*|*39.41*|0.9909|
> |Tra x 3|230.28|19.01|37.95|0.9908|
> |Alt x 2|230.58|20.25|38.02|0.9908|
> |Alt x 3|232.25|19.89|37.80|0.9907|
> |our AICT|*228.43*|17.76|39.40|**0.9910**|
>
> ***Q3: Explain why foreground-normalized MSE loss [34] is more important***
> ***A3:*** The foreground-normalized MSE loss, widely used in methods like PCT-Net and DCCF, prevents instability in images with very small objects. For ablation studies, we set Amin to 100, 5000, and 10000, denoted as 'Amin=100', 'Amin=5000', and 'Amin=10000'. In our AICT, Amin is set to 1000. As shown in the table, when Amin is 100, performance doesn't improve due to the scarcity of smaller targets. Increasing Amin from 1000 to 10000 causes instability in training on small targets, resulting in decreased performance.
>
> |Method|fMSE↓|MSE↓|PSNR↑|SSIM↑|
> |----|:----:|:----:|:----:|:----:|
> |MSE|287.04|23.17|38.41|0.9908|
> |Amin=100|229.87|17.96|39.23|0.9909|
> |Amin=5000|234.22|20.33|38.69|0.9852|
> |Amin=10000|250.89|22.56|37.56|0.9864|
> |our AICT|**228.43**|**17.76**|**39.40**|**0.9910**|
>
> ***Q5: Report non-reference image quality metrics (e.g., NIQE and LPIPS) for Figure 9.***
> ***A5:*** Since LPIPS is a full-reference metric, we only report NIQE in the table below. Our method obtains the best results in the first two examples and the second-best result in the last example.
>
> |Method|Composite image|Harmonizer|DCCF|PCT-Net|our AICT|
> |----|:----:|:----:|:----:|:----:|:----:|
> |NIQE↓|10.51|9.18|*9.09*|9.35|**9.06**|
> |NIQE↓|2.85|*2.82*|2.82|2.82|**2.82**|
> |NIQE↓|3.21|3.15|3.16|**3.15**|*3.15*|
>
> ***Q6: Provide intermediate visual results of the ablated models***
> ***A6:*** The intermediate visual results are shown in Figure 4 of the PDF attached in the author rebuttal.
>
> ***Q7: Explain why the channel-crossing strategy fails to improve performance***
> ***A7:*** Our parameter model uses an RGB image to predict parameter maps, which inherently include information from all color channels. The channel-crossing strategy introduces redundancy without improving performance.
>
> ***Q8: Use SSIM as the evaluation metric***
> ***A8:*** SSIM is actually not a commonly used metric for image harmonization tasks. Following this suggestion, we have used SSIM as an evaluation metric. As shown in the table, our AICT achieves a competitive SSIM score of 0.9910.
>
> |Method|Harmonizer|DCCF|PCT-Net|our AICT|
> |----|:----:|:----:|:----:|:----:|
> |SSIM↑|0.9685|0.9896|**0.9911**|0.9910|

---

> > ### Comment · Reviewer_M5vB · 2024-08-08
> >
> > Thanks and I have read the rebuttal, which has provided enough information.

---

> > > ### Author Response · Authors · 2024-08-09
> > >
> > > We are very happy that our rebuttal provides enough information for addressing your concerns. It would be very appreciated if you could increase your rating if our rebuttal has addressed your concerns.  Thank you very much for your efforts in reviewing our paper.

---

### Official Review · Reviewer_4Tt1 · 2024-06-16

**Soundness:** 3
**Presentation:** 3
**Contribution:** 3
**Rating:** 7
**Confidence:** 5

**Summary:**

This paper proposes an AdaptiveInterval Color Transformation method (AICT) for high-resolution image harmonization, which predicts pixel-wise color transformation and adaptively adjusts the sampling interval to model local non-linearities of the color transformation at high resolution.

**Strengths:**

1. The paper is overall well-written and easy to understand.
2. The idea is neat and clean.
3. The experimental results look good.

**Weaknesses:**

1. The authors should conduct experiments using the training/test set of HAdobe5k which focuses on high-resolution, and report the results under different resolutions (1024, 2048, etc). More baselines should be compared in this setting.
2. The authors should test on high-resolution real composite images and conduct user study.
3. The authors are suggested to evaluate the method on ccHarmony dataset, which can reflect the illumination variation more faithfully.
4. The authors should discuss the limitation of the proposed method and show some failure cases.
5. More efficiency analyses (FLOPs, memory, time) should be provided.

**Questions:**

Please address the questions in "Weakness".

**Limitations:**

See the "Weakness".

---

> ### Author Rebuttal · Authors · 2024-08-06
>
> **Thanks for your positive evaluation and valuable suggestions.**
>
> ***Q1: The authors should conduct experiments using the training/test set of HAdobe5k which focuses on high-resolution, and report the results under different resolutions (1024, 2048, etc). More baselines should be compared in this setting.***
> ***A1:*** Thanks for this suggestion. We have reported the results under the  resolutions 1024x1024 and 2048x2048 using the training/test set of HAdobe5k in Tables 2 and 7 of the submitted manuscript. Following this suggestion, we present comparison with two additional baselines:  AdaInt [1] and SepLUT [2]. As shown in the tables below, our method outperforms AdaInt and SepLUT in terms of all metrics.
>
> |Image Size|Method|fMSE↓|MSE↓|PSNR↑|SSIM↑|
> |----|:----:|:----:|:----:|:----:|:----:|
> |1024 x 1024|AdaInt|235.86|31.75|37.80|0.9837|
> ||SepLUT|224.98|27.41|37.75|0.9826|
> ||Our AICT|**156.81**|**19.50**|**39.67**|**0.9863**|
>
> |Image Size|Method|fMSE↓|MSE↓|PSNR↑|SSIM↑|
> |----|:----:|:----:|:----:|:----:|:----:|
> |2048 x 2048|AdaInt|221.90|31.09|38.10|0.9846|
> ||SepLUT|216.14|27.02|37.97|0.9834|
> ||Our AICT|**147.99**|**17.92**|**40.07**|**0.9871**|
>
> [1] Yang, Canqian, et al. AdaInt: Learning adaptive intervals for 3D lookup tables on real-time image enhancement. CVPR 2022.
>
> [2] Yang, Canqian, et al. Seplut: Separable image-adaptive lookup tables for real-time image enhancement. ECCV 2022.
>
> ***Q2: The authors should test on high-resolution real composite images and conduct user study.***
> ***A2:*** Thanks for this suggestion. We test on a high-resolution real composite image dataset [3] with resolutions ranging from 1024 × 1024 to 6016 × 4000. We randomly select 20 images for the user study. In the study, 20 volunteers independently rank the predictions from 1 to 3 based on visual quality, considering color and luminance consistency. Scores of 3, 2, and 1 are assigned for ranks 1, 2, and 3, respectively. The mean scores for each method are presented in the table below. As we can see that our method achieves the highest score.
>
> |Method|Harmonizer|DCCF|PCT-Net|our AICT|
> |----|:----:|:----:|:----:|:----:|
> |Score↑|21.6|28|32.4|**37.8**|
>
> [3] Li Niu, et al. Deep image harmonization with globally guided feature transformation and relation distillation. ICCV 2023.
>
> ***Q3: The authors are suggested to evaluate the method on ccHarmony dataset, which can reflect the illumination variation more faithfully.***
> ***A3:*** Thanks for this suggestion. Actually, we have evaluated the method on the ccHarmony dataset and presented the results in Table 3 of the submitted manuscript. Our method achieves the best results in terms of MSE, PSNR, and SSIM metrics, which demonstrates the effectiveness of the proposed method when handling illumination variation.
>
> ***Q4: The authors should discuss the limitation of the proposed method and show some failure cases.***
> ***A4:*** Thanks for this suggestion. We have discussed the limitations of the proposed method and shown some failure cases in Section A.4 of the supplementary materials of the submitted paper.
>
> ***Q5: More efficiency analyses (FLOPs, memory, time) should be provided.***
> ***A5:*** Thanks for this suggestion. In the submitted manuscript, we have provided the FLOPs, memory cost, and inference time on the HAdobe5k dataset for different resolutions (1024 x 1024 and 2048 x 2048). Please refer to Tables 8 and 9 of the submitted manuscript.

---

> > ### Comment · Reviewer_4Tt1 · 2024-08-09
> >
> > The rebuttal has addressed all my concerns.

---

> > > ### Author Response · Authors · 2024-08-09
> > >
> > > We are very happy that our rebuttal has addressed all your concerns.  Thank you very much for your efforts in reviewing our paper.

---

### Official Review · Reviewer_dAiu · 2024-07-05

**Soundness:** 3
**Presentation:** 2
**Contribution:** 3
**Rating:** 5
**Confidence:** 4

**Summary:**

This paper presents a new method for harmonizing the color of foreground objects added to scenes with the colors of the background original image. The authors focus on a model able to present good results on high-resolution images. The idea is to learn two different sets of Look-Up-Tables. The first set aims at modifying the input values to re-scale them in a way that the second set can perform pixel-wise editing. The authors show the ability of their method in three different datasets.

**Strengths:**

- The general idea of using two LUT-like transformations is interesting.
- Results outperform the state of the art.

**Weaknesses:**

- The paper is difficult to follow at the beginning, as the authors abuse notation regarding 3DLUTs in color image processing. In color image processing 3DLUTs are a function that goes from R^3 to R^3, i.e. given an input (R,G,B) values they output a (R',G',B') value ---see for example [1]---. Therefore, in color image processing 3DLUTs cannot be approximated by curves (contrary to what is said by the authors in lines 138-139). What the authors are doing is learning spatial-aware 1D LUTs, i.e. a function from R^3 to R that given an input (x,y,R) outputs a value R'. It is true that some papers (citation 24 on the paper) already abuse notation when speaking of 4DLUTs. However, in their case, they are not redefining an already existing term, as it is 3DLUTs. Therefore, the authors of this manuscript need to rewrite different parts of the paper refraining from using the term 3DLUT to define their approach. As it currently stands, it cannot be accepted due to this conceptual error.

- Metrics are insufficient. MSE and PSNR are strongly correlated metrics. In the end, PSNR=10log_{10}(MAX^2/MSE), where MAX is the maximum possible value. Therefore, the authors need to add further metrics such as LPIPS, DeltaE as perceptually-based full-referenced metrics, and NIQE as a non-reference one.

- The paper will be of much larger interest if the authors present the results of a user study, for example as it was done in PCT-Net (reference 12 in this paper) or in [2].

- Figures 1,3, and 4 are versions of the same general idea of Figure. It will be beneficial for the paper if the authors can present just 1 Figure in which all the small differences are included.

[1] Conde, M. V., et al. Nilut: Conditional neural implicit 3d lookup tables for image enhancement, AAAI 2024
[2] Valanarasu, J. M. J., et al. (2022). Interactive portrait harmonization, ICLR 2023

**Questions:**

First, please address the weaknesses mentioned before.

Also, when reading this paper, I got the feeling that: "This idea is mostly AdaInt + Mask". I may have lost some important component here, so please guide me through why my feeling may not be correct.

**Limitations:**

Yes

---

> ### Author Rebuttal · Authors · 2024-08-06
>
> **Thanks for your positive evaluation and valuable suggestions.**
>
> ***Q1:The authors abuse notation regarding 3DLUTs in color image processing.***
> ***A1:*** We thank the reviewer very much for pointing out the inaccurate usage of the term 3DLUTs. Following the reviewer's suggestion,  we will rename the term of 3DLUTs  to define our approach and rewrite the corresponding parts of the paper.
>
> ***Q2: Metrics are insufficient. The authors need to add further metrics such as LPIPS, and DeltaE as perceptually-based full-referenced metrics, and NIQE as a non-reference one.***
> ***A2:*** Thanks for this suggestion. Following this suggestion, we use LPIPS, DeltaE, and NIQE metrics to evaluate the  performance of the compared methods on the  iHarmony4 dataset. The table below shows the quantitative results.   As we can see that our method achieves the best results in terms of LPIPS and DeltaE and the third best results in terms of NIQE. It should be noted that NIQE is a no-reference metric that does not assess color or semantic context continuity. Therefore, it is not suitable for evaluating the performance of image harmonization.
>
> |Method|LPIPS↓|DeltaE↓|NIQE↓|
> |----|:----:|:----:|:----:|
> |Harmonizer|0.016|0.88|4.48|
> |DCCF|0.017|0.80|**4.46**|
> |PCT-Net|**0.013**|*0.67*|*4.47*|
> |our AICT|**0.013**|**0.66**|4.48|
>
> ***Q3: The paper will be of much larger interest if the authors present the results of a user study, for example as it was done in PCT-Net (reference 12 in this paper) or in [2].***
> ***A3:*** Thanks for this suggestion. We randomly select 20 high-resolution real composite images from the GiftNet dataset. In the study, 20 volunteers independently rank the predictions from 1 to 3 based on visual quality, considering color and luminance consistency. Scores of 3, 2, and 1 are assigned for ranks 1, 2, and 3, respectively. The mean scores for each method are presented in the table below. As we can see our method achieves the highest score.
>
> |Method|Harmonizer|DCCF|PCT-Net|our AICT|
> |----|:----:|:----:|:----:|:----:|
> |Score↑|21.6|28|32.4|**37.8**|
>
> ***Q4: Figures 1,3, and 4 are versions of the same general idea of Figure. It will be beneficial for the paper if the authors can present just 1 Figure in which all the small differences are included.***
> ***A4:*** Thanks for this suggestion. We will combine Figures 1, 3, and 4 into a single comprehensive figure that highlights all the small differences in the camera-ready version.
>
> ***Q5: Also, when reading this paper, I got the feeling that: "This idea is mostly AdaInt + Mask". I may have lost some important component here, so please guide me through why my feeling may not be correct.***
> ***A5:*** We are sorry that our presentation let the reviewer misunderstand that our idea is mostly AdaInt + Mask. In fact,  our method is significantly distinct from AdaInt in three aspects. **Firstly**, AdaInt predicts only a few weight parameters to fuse several fixed LUTs obtained through training, which limits its expressive capacity. In contrast, our method dynamically predicts entire LUTs based on the input, resulting in significantly greater expressive power. **Secondly**, AdaInt uses LUTs to perform global RGB-to-RGB transformations, which overlook diverse local contexts. In contrast, our method uses the color and position of each pixel as inputs for the LUTs to perform pixel-wise color transformations. **Thirdly**, AdaInt applies adaptive interval learning to each lattice dimension of the 3D LUTs for global sampling interval adjustment, which limits the expressiveness of the 3D LUTs. In contrast, our method achieves pixel-wise sampling interval adjustment to model local non-linearities of the color transformation.

---

> > ### Comment · Reviewer_dAiu · 2024-08-09
> > **Increasing my rating**
> >
> > Following the answer, and although results in LPIPS, Delta E and NIQE are not impressive, I increase my rating to a borderline accept in line with the other reviewers.
> >
> > This said, please rename the term 3DLUT in the paper as promised in the rebuttal.

---

> > > ### Author Response · Authors · 2024-08-10
> > >
> > > Thank you very much for your efforts in reviewing our paper. We will certainly rename the term 3DLUT in the paper as promised in the rebuttal.

---

### Official Review · Reviewer_pwQP · 2024-07-13

**Soundness:** 3
**Presentation:** 2
**Contribution:** 2
**Rating:** 6
**Confidence:** 4

**Summary:**

This paper proposes a new method called Adaptive-Interval Color Transformation (AICT) for high-resolution image harmonization. The key ideas are:

1. Predicting pixel-wise color transformations using a parameter network that generates multiple 3D lookup tables (LUTs).
2. Separating the color transform into cascaded sub-transformations using two 3D LUTs to adaptively adjust the sampling intervals and model local non-linearities.
3. Using a global consistent weight learning module to predict image-level weights for each transformation to enhance overall harmony.

Extensive experiments show that AICT achieves state-of-the-art performance on benchmark datasets while being computationally efficient.

**Strengths:**

1. **Novel approach to high-resolution image harmonization**: The paper proposes a new method, Adaptive-Interval Color Transformation (AICT), which addresses the challenging problem of harmonizing high-resolution composite images. The key idea of using adaptive sampling intervals in the color transformation is a novel contribution that sets AICT apart from prior works.

2. **Improved quantitative performance on benchmark datasets**: AICT achieves state-of-the-art results on the widely-used iHarmony4 and HCOCO datasets, outperforming previous methods in terms of fMSE, MSE, and PSNR metrics. These quantitative improvements, although relatively small in some cases, demonstrate the effectiveness of the proposed technical innovations.

3. **Computational efficiency and scalability**: The paper presents a detailed analysis of the computational complexity and memory usage of AICT, showing that it achieves a favorable trade-off between performance and efficiency compared to existing high-resolution harmonization methods. The scalability of AICT to higher resolutions (e.g., 2048x2048) is also demonstrated, which is important for practical applications.

4. **Insights into the role of adaptive sampling intervals**: Through ablation studies and visualizations, the paper provides some insights into the importance of adaptive sampling intervals in the color transformation process. The comparison between fixed and adaptive intervals highlights the benefits of the proposed approach in terms of handling local color variations and edge artifacts.

**Weaknesses:**

## Unrealistic Settings
The input resolutions in the main experiments (1024x1024 and 2048x2048) are still relatively low compared to professional compositing workflows, which often deal with 4K or higher resolutions. The effectiveness of AICT for ultra high-resolution images is unclear and should be validated with appropriate benchmarks.

## Lack of Comparison to Extensively Studied Prior Art: Proposed Approach Appears to be a Trivial Amalgamation of Established Techniques
- The use of 3D LUTs for color transformations is not a fundamentally new idea in image processing. Many previous works, especially in the photo enhancement domain, have used similar techniques. The paper does not sufficiently explain how AICT's specific LUT-based approach is distinct from prior methods.
- Adaptively adjusting the sampling intervals of color transformations has also been explored in works like AdaInt. While AICT's dual cascaded LUTs are a nice extension, this still seems like an incremental contribution building on existing ideas.
- The global weight learning module likewise appears very similar to the global adjustment parameters used in existing works. More discussion is needed on how AICT's weights differ and what additional capabilities they enable.

## Missing motivations for method modules

- The global weight learning module seems disconnected from the rest of the method. It's unclear why weighting the predicted LUTs based on global image features would improve harmonization quality. The authors should explain the reasoning behind this design choice and how it relates to the adaptive interval learning.
- The global weight learning module predicts a single scalar weight for each LUT, which is then uniformly applied to all pixels. This global approach contradicts the motivation of AICT to enable local color adjustments. A more spatially-varying weighting scheme would be more consistent with the overall goal of the method.


## Evaluation

- The use of cascaded dual 3D LUTs is a key aspect of AICT, but is not ablated. Testing a single LUT baseline, more numbers of LUTs or alternative compositions of LUTs would help validate the necessity of the proposed decomposition.
- The user study results are not reported, making it difficult to assess AICT's perceptual quality compared to other methods. Quantitative metrics like fMSE and PSNR do not always correlate well with human judgments, so user ratings would provide valuable additional evidence for AICT's benefits.

## Performance
While the method outperforms previous approaches, the gains on some metrics (e.g. PSNR) are relatively modest (<1 dB). It would be good to see more analysis on what types of images/scenarios benefit most from AICT.


Overall, the reviewer acknowledges that this paper addresses an important topic using generally reasonable methods. The initial score will be increased if the rebuttal effectively addresses the concerns mentioned above.

**Questions:**

Please see weakness.

**Limitations:**

Please see weakness.

---

> ### Author Rebuttal · Authors · 2024-08-06
>
> **Thanks for your positive evaluation and valuable suggestions.**
>
> ***Q1: The effectiveness of AICT for ultra high-resolution images is unclear and should be validated with appropriate benchmarks.***
> ***A1:*** Thanks for this suggestion. In Table 1 of the paper, we have reported the results  on the HAdobe5k  dataset with resolutions ranging from 312 × 230 to 6048 × 4032. Following the reviewer's suggestion, we collect two ultra high-resolution benchmarks (over4K and over5K) by selecting images with resolutions exceeding 4K (4096 × 2160) and 5K (5120 × 2880) from the HAdobe5k dataset. The compared results on the table below show that our method achieve the best performance on the over4K benchmark in terms of all metrics.  On the over5K benchmark, our method achieves the best performance in terms of MSE and PSNR and the second-best performance in terms of fMSE and SSIM.
>
> |Benchmark|Method|fMSE↓|MSE↓|PSNR↑|SSIM↑|
> |----|:----:|:----:|:----:|:----:|:----:|
> |  over4K  |Harmonizer|199.68|22.87|37.91|0.9374|
> ||DCCF|199.85|20.54|38.38|0.9884|
> ||PCT-Net|165.31|19.36|39.82|**0.9900**|
> ||AdaInt|226.66|27.83|38.20|0.9882|
> ||SepLUT|219.60|24.98|38.13|0.9866|
> ||Our AICT|**154.44**|**16.79**|**40.16**|**0.9900**|
>
> | benchmark|Method|fMSE↓|MSE↓|PSNR↑|SSIM↑|
> |----|:----:|:----:|:----:|:----:|:----:|
> |over5K|Harmonizer|302.43|35.09|36.65|0.9441|
> ||DCCF|320.02|28.44|35.73|0.9757|
> ||PCT-Net|**292.62**|*21.40*|*37.48*|**0.9790**|
> ||AdaInt|330.25|29.26|36.35|0.9781|
> ||SepLUT|395.42|22.06|36.30|0.9774|
> ||Our AICT|*293.08*|**15.48**|**37.64**|*0.9786*|
>
> ***Q2 & Q3: Explain how AICT's specific LUT-based approach is distinct from prior methods.***
> ***A2 & A3:*** Compared with prior methods, our method has three distinct differences. **Firstly**, unlike previous methods that use fixed weight parameters to fuse pre-trained LUTs, our method dynamically predicts entire LUTs based on the input. **Secondly**, Previous methods use LUTs for global RGB-to-RGB transformations, which often overlook local context. In contrast, our method uses each pixel’s color and position as inputs for LUTs, enabling pixel-wise color transformations. **Thirdly**, previous methods apply adaptive interval learning and separable lookup tables to each lattice dimension of the 3D LUTs for global sampling interval adjustment, which limits the expressiveness of the 3D LUTs. In contrast, our method achieves pixel-wise sampling interval adjustment to model local non-linearities of the color transformation.
>
> ***Q4 & Q5 & Q6: More discussion is needed on how AICT's weights differ and what additional capabilities they enable.***
> ***A4 & A5 & A6:*** Thanks for this suggestion. When a foreground is inserted into a new background, its average brightness is influenced by the background lighting. Predicting spatially consistent weights for global color adjustment can enhance adaptive interval learning. To further discuss AICT's weights, we conduct ablation studies by sequentially removing the parameters: first for the R channel ('w/o R'), then for both R and G channels ('w/o RG'), and finally all parameters ('w/o weight'). We also explore pixel-level weights ('spatial'). As shown in the first table below, increasing the number of removed parameters decreases performance, highlighting the role of weight learning module. The results of "spatial" show that learning spatially-varying weights does not enhance performance, as LUTs handle pixel-wise color transformations. To illustrate the role of the weight learning module in global color adjustment, we calculate the error between the average color values of harmonized and ground truth images. As shown in the second table below, our method performs better.
>
> |Method|fMSE↓|MSE↓|PSNR↑|SSIM↑|
> |----|:----:|:----:|:----:|:----:|
> |w/o R|230.83|18.25|39.39|**0.9910**|
> |w/o GB|232.20|19.10|39.18|0.9909|
> |w/o weight|247.48|19.40|39.16|0.9908|
> |spatial|230.24|18.03|39.23|0.9909|
> |our AICT|**228.43**|**17.76**|**39.40**|**0.9910**|
>
> |Method|w/o weight|our AICT|
> |----|:----:|:----:|
> |error↓|6.94|**6.57**|
>
> ***Q7: Cascade LUTs need ablation studies***
> ***A7:*** Following this suggestion, we report more results in the table below. For each color channel, we use only 1 LUT for color transformation ('Single') and cascade 2 and 3 LUTs for adaptive interval learning ('Int x 2', 'Int x 3') and color transformation ('Tra x 2', 'Tra x 3'). We also cascade 4 and 6 LUTs for alternating adaptive interval learning and color transformation ('Alt x 2', 'Alt x 3'). Performance improves slightly with more LUTs but plateaus at three, due to increased training difficulty.
>
> |Method|fMSE↓|MSE↓|PSNR↑|SSIM↑|
> |----|:----:|:----:|:----:|:----:|
> |Single |248.51|19.71|39.15|0.9908|
> |Int x 2|227.95|**17.40**|**39.50**|**0.9910**|
> |Int x 3|229.50|18.55|38.95|0.9909|
> |Tra x 2|**228.40**|*17.50*|*39.41*|0.9909|
> |Tra x 3|230.28|19.01|37.95|0.9908|
> |Alt x 2|230.58|20.25|38.02|0.9908|
> |Alt x 3|232.25|19.89|37.80|0.9907|
> |our AICT|*228.43*|17.76|39.40|**0.9910**|
>
> ***Q8: The user study results are not reported.***
> ***A8:*** Thanks for this suggestion. We randomly select 20 high-resolution real composite images from the GiftNet dataset. In the user study, 20 volunteers independently rank the predictions from 1 to 3 based on visual quality. Scores of 3, 2, and 1 are assigned for ranks 1, 2, and 3, respectively. The mean scores for each method are presented in the table below. As we can see our method achieves the highest score.
>
> |Method|Harmonizer|DCCF|PCT-Net|our AICT|
> |----|:----:|:----:|:----:|:----:|
> |Score↑|21.6|28|32.4|**37.8**|
>
> ***Q9: More analysis on what types of images/scenarios benefit most from AICT.***
> ***A9:*** We present more results in Figure 2 in the PDF attached in the author rebuttal. As we can see that our AICT method particularly benefits images with rich textures, such as fur and hair. This is because AICT achieves non-uniform sampling intervals, allowing for better modeling of local non-linearities in color transformations.

---

> > ### Comment · Reviewer_pwQP · 2024-08-09
> >
> > Thank you for thoroughly addressing most of my concerns in your rebuttal. This has led me to increase my rating for the paper. I wish you the best of luck with your research!

---

> > > ### Author Response · Authors · 2024-08-09
> > >
> > > Thank you for your positive response to our rebuttal.  We appreciate all your efforts in reviewing our paper.

---

### Author Rebuttal · Authors · 2024-08-07

Dear Reviewers,

We would like to express our sincere gratitude to all the reviewers for their constructive feedback and for recognizing the performance and efficiency of our proposed method. We appreciate your valuable suggestions and have carefully addressed each point in our responses.

1.**Reviewer pwQP (Score: 5 - Borderline Accept)** recognizes the novel aspects of our AICT method and its computational efficiency. We have addressed your concerns by validating our approach on ultra high-resolution images and clarifying its distinctiveness from prior methods. Additionally, we have provided detailed explanations for the global weight learning module and included extensive ablation studies.

2.**Reviewer dAiu (Score: 4 - Borderline Reject)** acknowledges the innovative use of LUT transformations and the performance of our method compared to state-of-the-art techniques.  Following the reviewer's suggestion, we will rename the term of 3DLUTs to define our approach and rewrite the corresponding parts of the paper. We have added LPIPS, DeltaE, and NIQE metrics to our evaluation, demonstrating the superiority of our method. A user study has been conducted to further validate our results. Additionally, we have highlighted the key differences between our method and AdaInt to address your concerns.

3.**Reviewer 4Tt1 (Score: 7 - Accept)** acknowledges the clarity and the effectiveness of our method. In response to your suggestions, we have conducted additional experiments using the HAdobe5k dataset at various resolutions (1024x1024 and 2048x2048), compared our method with more baselines, and reported the results, demonstrating superior performance in terms of MSE and PSNR. We have also evaluated our method on high-resolution real composite images and the ccHarmony dataset, and discussed method limitations and failure cases in detail. Furthermore, we expand our efficiency analysis to include FLOPs, memory usage, and inference time for different resolutions.

4.**Reviewer M5vB (Score: 5 - Borderline Accept)** praises the neat and clean idea as well as the impressive experimental results. We have clarified the novelty of the 3D LUTs-based approach and demonstrated the advantages of AICT through comparative experiments. We address the issues related to non-linearities and efficiency, and adjust our claims accordingly. Additionally, we expand the discussion on the global consistent weight method, conduct further ablation studies, and correct errors and redundancies in the text.

Our responses to individual comments  of each reviewer are posted in the rebuttal under each reviewer's report.  All the required experimental results are presented in the PDF attached in  this rebuttal.

Specifically:

- **Figure 1**  visualizes the error between the harmonized and ground truth images.

- **Figure 2**  presents qualitative results against existing methods.

- **Figure 3**  displays the color distribution in local areas of high-resolution harmonized and ground truth images, showing that the pixel values predicted by our method are closer to the real color distribution.

- **Figure 4**  presents the visual results of  ablated models.

For convenience, we highlight the figure relevant to each reviewer's comments as follows:

- **Reviewer pwQP**: Figure 2.
﻿
- **Reviewer M5vB**: Figure 1, Figure 2, Figure 3, and Figure 4

We hope that our responses and the revisions made to the manuscript will alleviate any concerns and enhance the overall quality of our submission. Once again, we thank all the reviewers for their thoughtful reviews and valuable suggestions.

---

### Decision · Program_Chairs · 2024-09-25

**Decision:**

Accept (poster)

**Comment:**

This paper proposes a method for high-resolution image harmonization. It predicts pixel-wise color transformations using a parameter network that generates multiple 3D lookup tables (LUTs). It separates the color transform into cascaded sub-transformations using the LUTs. And a global consistency is proposed for the optimized performance. After the rebuttal, all the reviewers find their concerns have been well addressed and give positive recommendation for this submission.